# Tackling Modeling and Kinematic Inconsistencies by Fixed Point Iteration-Based Adaptive Control

Awudu Atinga [1,*,†,‡] and József K. Tar [1,2,3,†,‡]

1   Doctoral School of Applied Informatics and Applied Mathematics, Óbuda University,
    1034 Budapest, Hungary; tar.jozsef@nik.uni-obuda.hu
2   John von Neumann Faculty of Informatics, Óbuda University, 1034 Budapest, Hungary
3   Antal Bejczy Center of Intelligent Robotics, Óbuda University, 1034 Budapest, Hungary
*   Correspondence: ugn9n8.uni-obuda.hu@stud.uni-obuda.hu
†   Current address: Bécsi út 96/B, H-1034 Budapest, Hungary.
‡   These authors contributed equally to this work.

**Abstract:** The Fixed Point Iteration-based Adaptive Control design methodology is an alternative to the Lyapunov function-based technology. It contains higher-order feedback terms than the standard resolved acceleration rate control. This design approach strictly separates the kinematic and dynamic issues. At first, a purely kinematic prescription is formulated for driving the components of the tracking error to zero. Then an available approximate dynamic model is used to calculate the approximated necessary control forces. Before exerting on the controlled system, these forces are adaptively deformed in order to precisely obtain the prescribed kinematic behavior. The necessary deformation is iteratively found by the use of a contractive map that results in a sequence that converges to the unique fixed point of this map. In the case of underactuated systems, when the relative order of the control task also increases, the highest-order time-derivative depends on the lower-order ones according to the dynamic model of the system. This makes it impossible to realize the arbitrarily constructed kinematic design. In the paper, a resolution to this discrepancy is proposed. The method is demonstrated using two non-linear paradigms, a three-degree-of-freedom robot arm, and a two-degree-of-freedom system, i.e., two coupled non-linear springs. The operation of the method was investigated via simulations made by the use of Julia language and simple sequential programs. It was found that the suggested solution could be considered as a new variant of the fixed point iteration-based model reference adaptive control that is applicable for underactuated systems even if the relative order of the task is increased.

**Keywords:** adaptive control; fixed point iteration-based adaptive control; adaptivity by abstract rotations; Banach's fixed point theorem; model reference adaptive control

## 1. Introduction

Using the humble capacities of the available electronic devices, in the 1980s, as a "model-based approach", the idea of *"Computed Torque Control (CTC)"* was developed for robots [1,2]. To remove the mathematical complexities of the *"Model Predictive Control (MPC)"* that generally uses the dynamic model in the constraint terms of a complicated optimization task that worked well for slow chemical processes (e.g., [3–5]), in this approach the dynamic model is directly used for computing the necessary control torque or force components.

However, it became clear very early that, practically, it is impossible to develop precise enough dynamic models for robots (e.g., [6]). A reasonable control quality can be achieved by developing adaptive controllers that, based on real-time observations, compensate for the consequences of the imprecise models. A group of adaptive controllers tried to "learn the parameters of the exact models" (the appropriate prototypes are the *"Slotine-Li Adaptive Robot Controller"* and the *"Adaptive Inverse Dynamics Controller (AIDC)"* [7]). Instead of parameter tuning, the *"Model Reference Adaptive Controller (MRAC)"* approach introduced

fast feedback signals into the controlled system in order to make its dynamic behavior similar to that of a linear time-invariant *"reference system"* because it is easy to control such systems (early examples are, e.g., [8,9]). In the relatively fresh book chapter [10], it is stated that *"Lyapunov's direct method is introduced as an indispensable tool for analyzing the stability of nonlinear systems."* Really, the mainstream of designing adaptive controllers goes back to Lyapunov's second method he elaborated in his Ph.D. dissertation in 1892 [11], which became well-known by the Western world only in the 1960s [12]. It is generally used in various adaptive solutions (e.g., [13,14]).

The Lyapunov function-based technique is prevalent, but it has some drawbacks.

1.  The Lyapunov function-based design is not a simple algorithm that can be learned. It requires creative, mathematically well-educated designers. From the failure to find the appropriate Lyapunov function, no conclusions can be drawn regarding the solvability of the given problem, though for a large class of problems, appropriate *Lyapunov function candidates* are suggested for use (e.g., [15]).

2.  It usually guarantees normal or asymptotic convergence of a scalar norm made of various error components that individually have physical interpretations and significance and should be driven to zero monotonically. However, the various components of this composite norm do not converge to zero monotonically, even if this norm itself monotonically vanishes.

3.  Normally, complicated model terms must be precisely computed or at least estimated when this method is used.

In 2009, an alternative adaptive approach was initiated to tackle the above problems [16]. It can also be referred to as *"Fixed Point Iteration-based Adaptive Control (FPIAC)"*. Finding the appropriate control signal was transformed into iteratively computing the fixed point of the controlled system's *"Response Function"*. This fixed point iteration-based approach individually keeps control of the selected error components by realizing some kinematically prescribed time-dependence for them.

In control technology, besides the lack of precise dynamic models, underactuation often causes practical problems. In a comprehensive survey of the underactuated mechanical systems [17], underactuation is defined as *"An underactuated mechanical system (UMS) is a system which has fewer independent control actuators than degrees of freedom to be controlled."* (This concept, naturally, can be applied to a wider set of physical systems than Classical Mechanical ones). In addition to providing typical mechanical examples, this review classifies the underactuated systems according to the reasons for underactuation and system constraints (e.g., it considers holonomic and non-holonomic systems) by certain configuration characteristics and according to the related control problems. According to the state of the art at the time, the article mentioned the following methods which intended to control UMSs, including *"Partial Feedback Linearization (PFL)"* [18]; *"Collocated PFL"* [19]; *"Non-collocated PFL"* [20]; *"Passivity-based Control (PBC)"*, that is mainly used for setpoint regulation that is a narrow range of applications (e.g., for two-link manipulators [21], Furuta pendulum [22], and the so called TORA system [23]), its variants as *"Interconnection and Damping Assignment Passivity-based Control (IDA-PBC)"* [24], and the *"Controlled Lagrangian"* method [25]; *"Backstepping Control'* [26]; and *"Sliding Mode Control"* (e.g., [27], *"Fuzzy Control"* [28]). Due to its complexity, optimal control examples are not mentioned in our paper. A similar survey in 2018 concentrated on second-order underactuated systems [29], and extended the set of control methods with neural networks-based controllers [30].

The great majority of the above examples do not contain adaptive control. However, the adaptive ones as [28,31] are based on Lyapunov's technique. To improve this situation, the possibility of using fixed point iteration-based adaptivity for underactuated systems was investigated in the earlier works and is further developed in this paper by revealing and tackling certain modeling inconsistencies that formerly were not considered. The main benefit of this approach is that by the use of quite primitive "approximate system models", that may be realized by the use of simple embedded systems, quite complicated control tasks may be solved.

The paper is structured as follows: In Section 2, the closely related works are discussed. Section 3 contains the mathematically detailed problem formulation in two subsections depending on the existence of the need for developing a higher-order control approach in the control of the underactuated system. The simulation results and their noise sensitivity investigations are presented immediately after the description of the models. Section 4 summarizes the conclusions.

## 2. Related Work

In the previous works, the shortcomings of the Lyapunov function-based approach listed in Section 1 were addressed. For illustrating Shortcoming 1, i.e., mathematical complexity, **which is a simple structural issue**, the following paper provides a good example. In [32], a relatively simple system of the form of $\dot{x} = Ax(t) + bu(t) + bf(x(t), t)$ with state vector $x \in \mathbb{R}^n$, constant matrices with unknown elements $A \in \mathbb{R}^{n \times n}$, $b \in \mathbb{R}^{n \times 1}$, and an unknown bounded function $f(x, t)$ that represents the system non-linearities, model uncertainties, and the external disturbances was controlled. The authors provided a Lyapunov function-based solution using three special assumptions, five remarks, and two theorems, the mathematical details of which were expressed over pages 976–984 (approximately along 9 pages), and only the rest contained simulation results for a simple 2-degree-of-freedom system. Different Lyapunov functions were added in the complicated proof, which is a typical technique. The paper also serves as an example that normally, "observers" have to be developed for a Lyapunov function-based approach (Shortcoming 3).

In contrast to that, the fixed point iteration-based approach utilizes the fact that in a linear, normed, complete metric space (i.e., a Banach space) $\mathcal{B}$, the sequence generated by a *contractive* map $F : \mathcal{B} \mapsto \mathcal{B}$ so that $\{x_1; x_2 = F(x_1); \ldots; x_{i+1} = F(x_i); \ldots\}$ converges to the *unique fixed point of this function* $x_i \to x_\star = F(x_\star)$ as $i \to \infty$. The proof consists of a few simple lines.

By definition $F(x)$ is contractive if there exists $0 \le K < 1$ so that $\forall x, y \in \mathcal{B} \ \|F(x) - F(y)\| \le K\|x - y\|$. Since $\forall N \in \mathbb{N}$

$$
\begin{aligned}
&\|x_{n+N} - x_n\| = \|F(x_{n-1+N}) - F(x_{n-1})\| \le K\|x_{n-1+N} - x_{n-1}\| \le \ldots \\
&\le K^{n-1}\|x_{1+N} - x_1\| \to 0 \text{ as } n \to \infty \ .
\end{aligned}
\tag{1}
$$

By definition $\{x_n\}$ is a Cauchy sequence that in a complete space converges to a limit value $x_\star \in \mathcal{B}$. It is very easy to show that $x_\star$ is the fixed point of $F(x)$:

$$
\begin{aligned}
&\|F(x_\star) - x_\star\| = \|F(x_\star) - x_n + x_n - x_\star\| \le \|F(x_\star) - x_n\| + \|x_n - x_\star\| \le \\
&\le K\|x_\star - x_{n-1}\| + \|x_n - x_\star\| \to 0 \text{ as } n \to \infty \text{ since } x_n \to x_\star.
\end{aligned}
\tag{2}
$$

Finally, the uniqueness of $x_\star$ can be easily proved by the indirect manner. Assume that two different fixed points of function $F(x)$ exist as $y_\star \ne x_\star$ so that $F(y_\star) = y_\star$ and $F(x_\star) = x_\star$. From this it follows that

$$
\|y_\star - x_\star\| = \|F(y_\star) - F(x_\star)\| \le K\|y_\star - x_\star\|
\tag{3}
$$

That is a contradiction if $\|y_\star - x_\star\| > 0$. The only contradiction-free solution is $y_\star = x_\star$.

In the control applications, the solutions occur in the vicinity of the fixed point; therefore, it is enough if the contractivity is valid in this restricted region.

Regarding Shortcoming 2, the following can be said. In control technology, often *quadratic Lyapunov functions* are constructed from the individual error components in the form $V(t) = \|e(t)\|^2 = e(t)^T Pe(t)$ where $P$ is a constant symmetric positive definite matrix. It evidently may happen that $V(t)$ can decrease while certain components of the error $e(t)$ increase in absolute value at the cost of the decrease in other components. Since the Lyapunov function-based technique concentrates on asymptotically driving $V(t)$ to zero, this property has significance in the "initial transient phase" of the control when $V(t)$ is big;

consequently, there may be large error components whose further growth is undesirable. Evading this problem in the fixed point iteration-based solution again is a simple **structural issue**. Instead of prescribing the behavior of some function $V(t) \in \mathbb{R}$, the time-dependence of certain components $e_i(t)$ is directly controlled. (There is no need for using and computing any Lyapunov function).

In general, a monotonic decrease in the individual error components can be prescribed kinematically in various manners. The typical PID-type feedback results in fluctuation with decreasing amplitude, while various fractional order calculus-based techniques realize a monotonic decrease with the given sign of the error components. These techniques are prevalent in robotics (e.g., [33,34]). In [35], an excellent review can be found on the history of fractional order calculus. In [36], fractional order calculus inspired sequences were combined with the fixed point iteration-based technique. Furthermore, in [37], a simple approach was suggested to simulate lower order control strategies for higher order systems to evade non-monotonic fluctuation of the error terms; a simple proportional error decay rate was simulated for a second order system. A control technology-based tackling of treating patients suffering from *type 1 diabetes mellitus*, an early version of this adaptive control, was investigated via simulations [38].

For evading the full state estimation that normally is necessary for the Lyapunov function-based design (Shortcoming 3), the FPI-based technique was applied to adaptively control a two-degree-of-freedom system in [39]. This system consisted of a wheel and a mass-point. The wheel's rotational position was the observed and controlled variable. The system contained some coupled *"parasite dynamics"*; along one of the spokes of the wheel, a mass point placed between two springs was able to move. However, its position (i.e., the directly not controlled variable) and velocity were not observable. The suggested method was able to control the wheel's rotational motion between certain physically determined limits without complete state estimation. The appropriate details of the present simulation are discussed as the properties of the dynamic models used in this paper.

As a heuristic method, MPC is still popular because it can be combined with different approximations. To evade the use of complicated non-linear solvers in vehicle lateral control in [40], the dynamic model of the system was removed from the cost function. Only the limitations of these kinematic terms were deduced from the dynamic model that, in fact, was used only in an inner loop. This structure is similar to the FPI-based approach that also separates the kinematic and dynamic terms. It can be expected that this method, later, can be combined with the FPI-based adaptive technique.

In the following sections, the problem formulation and the FPI-based design structure are detailed.

## 3. Problem Formulation and Development of the Design Structure

Mathematically, the method worked based on Banach's Fixed Point Theorem [41], which was briefly summarized in Section 2. The structure of this controller is described in Figure 1 for a system in which the "order $\in \mathbb{N}$" time derivatives of the controlled coordinates the control force can instantaneously set. The method is designed for *digital controllers* in which the delay corresponds to the duration of the control cycle. In the *"Kinematic Block"*, various ideas can be applied that produce a *"desired"* $q^{(order)des}$ time-derivative that should be realized in order to drive the selected error components to zero.

It is evident that if the block *"Adaptive Deformation"* is removed from Figure 1, we arrive at the original CTC control. In the lack of the possession of a precise model, some available approximate dynamic models were used for calculating the necessary control force $Q(t)$ that is exerted on the controlled system, the response of which is the *realized time-derivative* $q^{(order)}(t)$. The function of the block called *"Adaptive Deformation"* is deforming the input

of the *"Approximate Model"* (denoted by $\tilde{\mathcal{M}}$) to achieve the case $q^{(order)des} = q^{(order)}$. This equation can be approximated as

$$Q = \tilde{\mathcal{M}}(q, \dot{q}, \ldots, q^{(order-1)}, q^{(order)des}) \tag{4a}$$

$$q^{(order)} = \mathcal{M}^{-1}(q, \dot{q}, \ldots, q^{(order-1)}, Q) \text{ leading to} \tag{4b}$$

$$q^{(order)} \cong \mathcal{M}^{-1}(\tilde{\mathcal{M}}(q^{(order)des})) = f(q^{(order)des}) \ , \tag{4c}$$

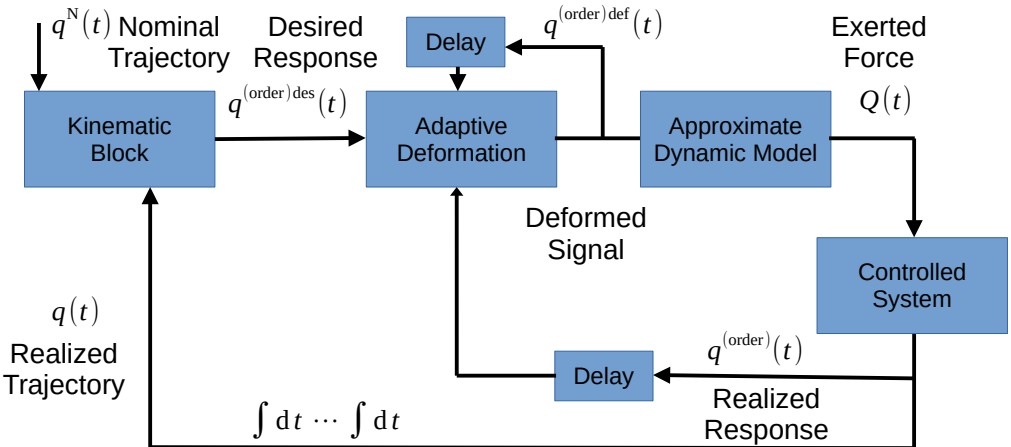

The FPI-based Adaptive Controller

**Figure 1.** The structure of the higher order Fixed Point Iteration-based Adaptive Controller for fully actuated system (after [16]).

It was taken into account that while the control force can immediately modify $q^{(order)}$, the other coordinate derivatives vary relatively slowly. In this manner, the *response function* $f(q^{(order)des})$ was introduced as an approximation. In practical applications, the *approximate dynamic model* $\tilde{\mathcal{M}}$ is known, but the *exact inverse model* $\mathcal{M}^{-1}$ is unknown. We should like to achieve the value $f(q^{(order)des})$. Assume we are near $x_\star$ that yields $f(x_\star) = x^{des}$, and let $\alpha$ be some real number. Try to use the iteration

$$x_{i+1} = x_i + \alpha \left( x^{des} - f(x_i) \right) \ ,$$

$$f(x_i) = f(x_\star + x_i - x_\star) \cong x^{des} + \frac{\partial f}{\partial x}(x_i - x_\star) \text{ from which it follows that} \tag{5}$$

$$x_{i+1} - x_\star \cong \left[ I - \alpha \frac{\partial f}{\partial x}\Big|_{x_\star} \right](x_i - x_\star) \ .$$

Evidently, if the matrix $\left[ I - \alpha \frac{\partial f}{\partial x}\Big|_{x_\star} \right]$ is *contractive*, the $x_i \to x_\star$ convergence can be guaranteed. Consider the variation of the norm of a transformed array $w$ as

$$\| [I - \alpha M] w \|^2 = \| w \|^2 - \alpha w^T (M + M^T) w + \alpha^2 w^T M^T M w \tag{6}$$

in which for small $\alpha$ the quadratic term can be neglected, so the second term must give a negative contribution. With the analogy of the monotonic function, the function $f : \mathbb{R}^n \mapsto \mathbb{R}^n$ can be regarded as *approximately differentially direction keeping* if for all $\Delta x$

$$\Delta x^T \Delta f = \Delta x^T (f(x + \Delta x) - f(x)) \cong \Delta x^T \frac{\partial f}{\partial x} \Delta x > 0, \tag{7}$$

that is, the angle between the vectors $\Delta x$ and $\Delta f$ is acute. Since any matrix, in our case $\frac{\partial f}{\partial x}$, can be decomposed into a symmetric and a skew symmetric part, and the latter does not give a contribution to $\Delta x^T \Delta f$ in Equation (7), it can be understood that in many applications convergence can be achieved if such a sequence is realized, e.g., in the box *"Adaptive Deformation"* with the inputs as follows, **the actual desired value** $x^{des} \equiv q^{(order)des}$; **the deformed value in the previous control cycle** $q^{(order)def}$; and the **the response obtained for the previously applied control input** $q^{(order)}$ that are available in the given time instant due to the delay. Because, in general, it is difficult to determine the appropriate value of parameter $\alpha$, various constructions were suggested for the deformation in [16,42,43]. They have various parameters by the setting of which the convergence of the iteration can be guaranteed if the system model allows it mathematically.

The most straightforward solution was announced in [44], which tries to move the response $f(x_i)$ toward $x_{i+1}^{des}$ in the following manner. At first, it augments the dimension of these vectors with additional physically not interpreted components so that the augmented vectors have the same Frobenius norm. It then constructs a rotation operator in the augmented space that rotates the augmented $f(x_i)$ into the augmented $x_{i+1}^{des}$ by leaving the vectors in their orthogonal subspace (this corresponds to a higher dimensional rotational axis) invariant. The similarly augmented version of $x_i$ is created and rotated with a fragment of the original rotation angle around the same rotational axis. Consequently, the physically interpreted projection of the rotated vector moves toward the desired direction, and, in this case, the interpolation factor $\lambda_a$ can be placed in the interval $[0, 1]$. Figure 2 intuitively describes the method. In [45], the original version published in [16] was combined with a genetic algorithm in order to optimize its parameters.

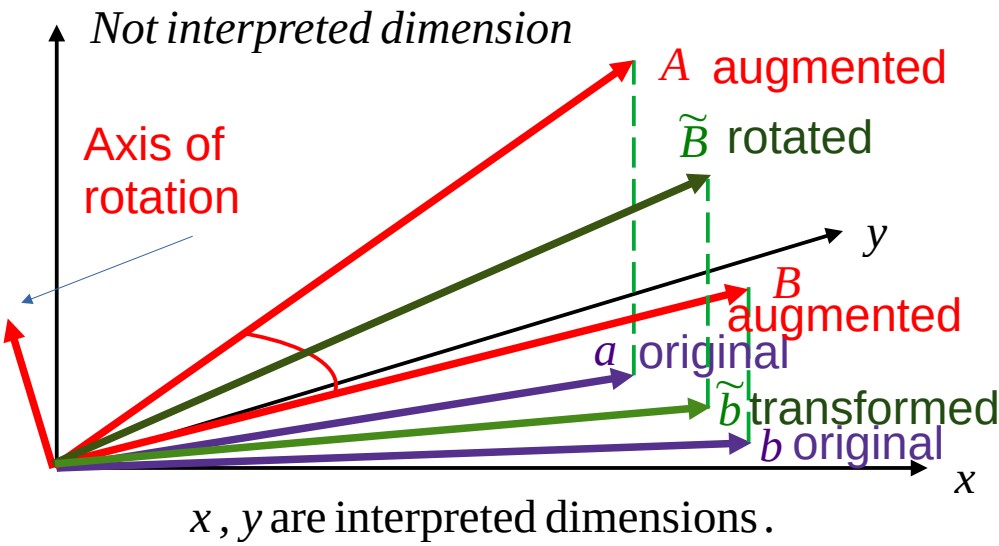

**Figure 2.** Symbolic description of the abstract rotations in the adaptive deformation (after [44]).

In [46], it was realized that if the adaptive iterative deformation is moved from the space of the derivatives to that of the generalized forces, a novel, *"Fixed Point Iteration-based Adaptive MRAC controller (FPI-based MRAC)"* can be developed for fully actuated systems as it is indicated in Figure 3. The purely kinematic design in the *"Kinematic Block"* continues the calculation with the dynamic data of the reference model. If the iteration converges in the space of the generalized forces, it has the illusion that the controlled system's dynamics is identical to that of the reference model. This latter can be a generally non-linear model, in contrast to the linear time-invariant reference model of the Lyapunov function-based design. For convergence, the same argumentation can be applied as in the case of the previous controller. The already systematically investigated applications of the above methods were made for fully actuated systems; however, in practice often can be found underactuated systems, the control of which is an exciting issue. The investigations

summarized in the sequel are the first systematic studies that reveal not only *modeling imprecisions* but also **modeling and kinematic inconsistencies** concerning the application of the fixed point iteration-based adaptive control for underactuated systems. These problems are revealed and tackled in the sequel.

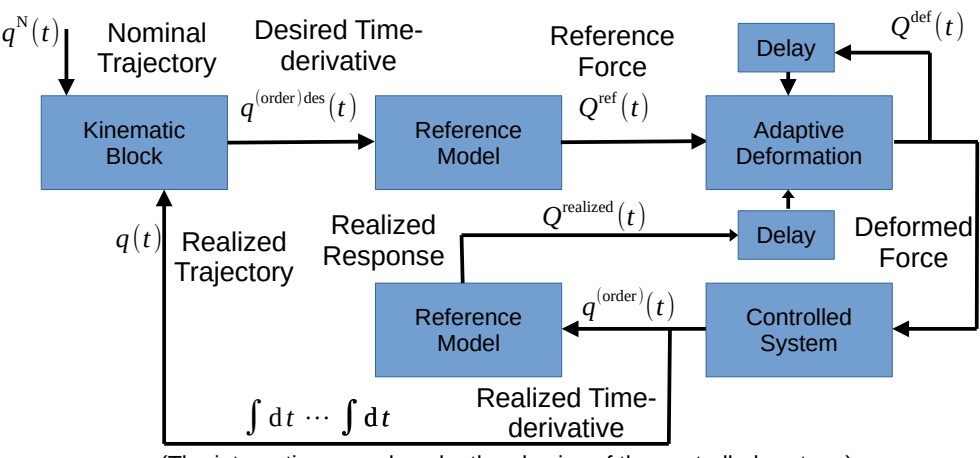

The FPI-based Model Reference Adaptive Controller for Full Actuation

**Figure 3.** The scheme of the Fixed Point Iteration-based MRAC controller for fully actuated systems (after [46]).

### 3.1. Suggested Controller Structures for Underactuated Systems without and with Increased Relative Order

Underactuation generally means that the number of independent control signals is smaller than that of the degree of freedom of the controlled system. Consequently, in this case, it is impossible to control the motion of each axle; specific axles will move *as they want*. However, the motion of a given directly not actuated axis can be controlled by the control force/torque exerted on a given directly actuated axis. It depends on the nature of the particular task if this solution will increase or not increase the *control task's relative order*. In this paper, both cases are investigated using two simple paradigms. The first is a robot arm in which the driver of one of the axles is corrupted and allows the appropriate axis to rotate freely. In this case, the relative order of the control is not increased. The other example consists of two linearly moving, dynamically coupled springs with mass points so that on the second (lower) mass point, no direct control force can be exerted. Its motion can be influenced by the force term directly acting on the first (upper) mass point, the motion of which is coupled to that of the lower mass point via viscous friction. In this case, the relative order of the control is increased from 2 to 3. This simple model represents a class of similar problems, e.g., in the *"Pneumatic Artificial Muscle (PAM)"* actuator, similar components are present that may cause oscillations (e.g., [47,48]). In [48], a method similar to the CTC control was applied for position control. If several pieces of the lower spring and mass systems are connected to the upper one so that, as parasite dynamics, they perturb its motion, the model of a multi-cantilever-mass mechanism can be developed. In [49], only linear springs were considered, and the main point was vibration suppression.

### 3.2. Control of a Corrupted 3D Puma-Type Robot Arm (Problem without Increasing the Relative Order of the Controller)

The dynamic model of the first three axles of a PUMA-type robot was investigated in [50], in this paper a relatively simple, corrupted version of this model will be used. Its kinematic structure is described in Figure 4.

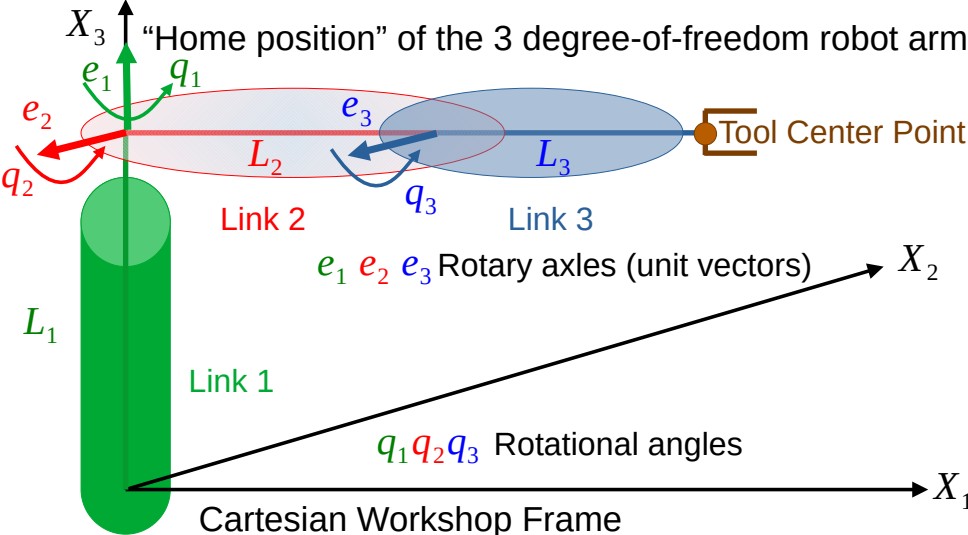

**Figure 4.** The kinematic structure of the 3D robot arm in the "home position" (after [50]).

It is assumed that the motion of the axles $q_1$ and $q_2$ can be directly controlled by the torque values $Q_1$ and $Q_2$, but the drive of the link of length $L_3$ is corrupted, therefore $Q_3 \equiv 0$. The following control compromise is applied, a *nominal trajectory to be tracked* is defined as $\dot{q}_1^N \equiv$ const., and $q_3^N(t) \equiv$ const. motion. The suggested control structure seems to be the modification of the FPI-based MRAC controller in which the available approximate model takes the role of the reference model (Figure 5). By the use of the $Q_3 \equiv 0$ equation for the approximate model $\ddot{q}_2^{des}$ can be computed and after that can be utilized for the calculation of $Q_1^{appr}$ and $Q_2^{appr}$. The dynamic and kinematic parameters of the robot arm are given in Table 1, the equations of motion (after [50]) are given in Equation (8).

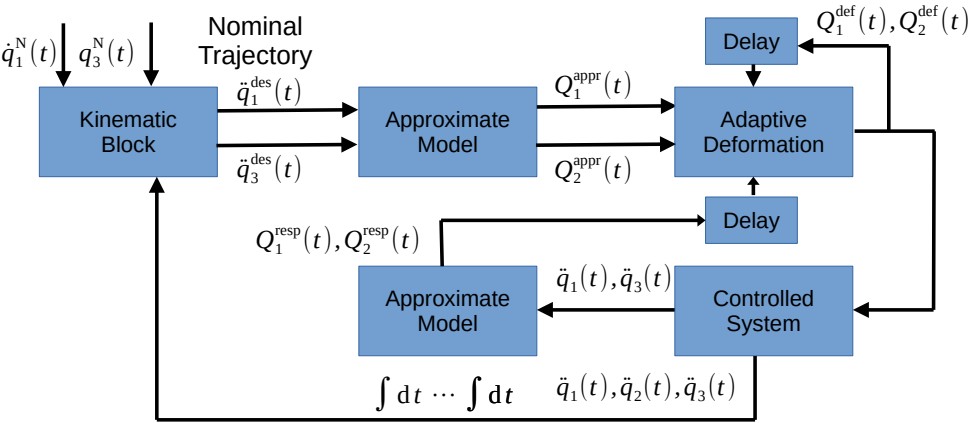

**Figure 5.** The control structure for the corrupted 3D robot arm.

**Table 1.** The kinematic and dynamic parameters of the robot arm (after [50]).

| Parameter | Measurement Unit | Exact Value | Approx. Value |
|---|---|---|---|
| Length of Link 2: $L_2$ | [m] | 1.0 | 1.0 |
| Length of link 3: $L_3$ | [m] | 2.0 | 2.0 |
| Mass of link 2: $m_2$ | [kg] | 10.0 | 15.0 |
| Mass of link 3: | [kg] | 20.0 | 25.0 |
| Gravitational acceleration $g$ | $[\mathrm{m \cdot s^{-2}}]$ | 9.81 | 9.81 |
| Moment of inertia of link 1: $\Theta$ | $[\mathrm{kg \cdot m^2}]$ | 50.0 | 60.0 |

$$H(q)\ddot{q} + h(q,\dot{q}) = Q \text{ , leading to} \tag{8a}$$

$$Q_1 = \left[\Theta_1 + \frac{1}{4}m_2 L_2^2 c_2^2 + \frac{1}{4}m_3 L_3^2 c_{23}^2 + m_3 L_2^2 c_2^2 + \frac{1}{2}m_3 L_2 L_3 c_{23} c_2\right]\ddot{q}_1 +$$

$$\left[-\frac{1}{2}m_2 L_2^2 c_2 s_2 \dot{q}_2 - \frac{1}{2}m_3 L_3^2 c_{23} s_{23}(\dot{q}_2 + \dot{q}_3) - 2m_3 L_2^2 c_2 s_2 \dot{q}_2\right.$$

$$\left. -\frac{1}{2}m_3 L_2 L_3 s_{23} c_2 (\dot{q}_2 + \dot{q}_3) - \frac{1}{2}m_3 L_2 L_3 c_{23} s_2 \dot{q}_2\right]\dot{q}_1 \text{ ,} \tag{8b}$$

$$Q_2 = \left[\frac{1}{4}m_2 L_2^2 + \frac{1}{4}m_3 L_3^2 + m_3 L_2^2 + \frac{1}{2}m_3 L_3 L_2 c_3\right]\ddot{q}_2 - \frac{1}{2}m_3 L_3 L_2 s_3 \dot{q}_3 \dot{q}_2$$

$$+ \left[\frac{1}{4}m_3 L_3^2 + \frac{1}{4}m_3 L_3 L_2 c_3\right]\ddot{q}_3 - \frac{1}{4}m_3 L_3 L_2 s_3 \dot{q}_3^2$$

$$+ \left[\frac{1}{4}m_2 L_2^2 c_2 s_2 + \frac{1}{4}m_3 L_3^2 c_{23} s_{23} + m_3 L_2^2 c_2 s_2 + \frac{1}{4}m_3 L_2 L_3 s_{23} c_2 + \frac{1}{4}m_3 L_2 L_3 c_{23} s_2\right]\dot{q}_1^2$$

$$+ \frac{1}{2}m_2 L_2 g c_2 + m_3 g L_2 c_2 + \frac{1}{2}m_3 L_3 g c_{23} \text{ ,} \tag{8c}$$

$$Q_3 = \left[\frac{1}{4}m_3 L_3^2 + \frac{1}{4}m_3 L_3 L_2 c_3\right]\ddot{q}_2 + \frac{1}{4}m_3 L_3^2 \ddot{q}_3 +$$

$$\left[\frac{1}{4}m_3 L_3^2 c_{23} s_{23} + \frac{1}{4}m_3 L_3 L_2 s_{23} c_2\right]\dot{q}_1^2$$

$$+ \frac{1}{4}m_3 L_3 L_2 s_3 \dot{q}_2^2 + \frac{1}{2}m_3 g L_3 c_{23} \text{ .} \tag{8d}$$

It has to be noted that not only the definition of the "home position of the robot" is different to that of [1], but the mass distribution of the components is different, too. Faitli used a simple "rod model" in which the masses were concentrated at half length of the rods. This model lead to $H_{12} = H_{21} = 0$, $H_{13} = H_{31} = 0$, and considerable $H_{11}$, $H_{22}$, and $H_{23} = H_{32}$ terms. Due to the more realistic mass distribution used, in [1] relatively little $H_{12} = H_{21}$ and $H_{13} = H_{31}$ terms occur, too. This difference does not concern the logic of the controller design.

For the kinematic block, if the tracking error is $e(t) := q^N(t) - q(t)$, and $e_{int}(t) := \int_{t_0}^t e(\xi)\mathrm{d}\xi$, the following tracking properties can be described with the *positive constants* $\Lambda_1$ and $\Lambda_3$:

$$\left(\Lambda_1 + \frac{\mathrm{d}}{\mathrm{d}t}\right)\dot{e}_1(t) \equiv 0 \text{ yielding } \ddot{q}_1^{des}(t) = \Lambda_1 \dot{e}_1(t) + \ddot{q}_1^N(t), \tag{9a}$$

$$\left(\Lambda_3 + \frac{\mathrm{d}}{\mathrm{d}t}\right)^3 e_{int3} \equiv 0 \text{ yielding } \ddot{q}_3^{des}(t) = \Lambda_3^3 e_{int3}(t) + 3\Lambda_3^2 e(t) + 3\lambda_3 \dot{e}(t) + \ddot{q}_3^N(t). \tag{9b}$$

The same iteration happens as in the case of the FPI-based MRAC controller. Physically it can be expected that via some fluctuation of $q_2(t)$, the axis $q_3(t)$ can be kept fluctuating around the constant nominal value prescribed for it. Due to the presence of gravity the existence of "static solution" cannot be expected.

To tackle the problem of measurement noises occurring in measuring the joint coordinates $q_1$, $q_2$, $q_3$, in the calculation of the control force, a simple low pass filter was used (even in the case of the noise-free simulations, too) in the following manner. The *measured/observed noisy value of the exact coordinate* $q(t)$ was $q^o(t) = q(t) + \mathcal{N}(t)$, in which the additional noise term corresponded to a random Gaussian noise of zero mean and $\sigma = 10^{-5}$ rad standard deviation. Instead of $q^o(t)$ and its numerical derivatives, the *smoothed value* $q^s(t)$ was utilized that satisfied the differential equation

$$\left( \lambda_s + \frac{\mathrm{d}}{\mathrm{d}t} \right)^3 q_i^s(t) = \lambda_s^3 q_i^o(t) \tag{10}$$

with the initial conditions $q_i^s(t_0) = q_i(t_0) = 0$, $\dot{q}_i^s(t_0) = \dot{q}_i(t_0) = 0$, and $\ddot{q}_i^s(t_0) = \ddot{q}_i(t_0) = 0$ with $\lambda_s = 10^3 \left[ \mathrm{s}^{-1} \right]$. Such a filtering may cause some delay even in the lack of the measurement noises.

The computation of the $Q_1^{appr}$ and $Q_2^{appr}$ force components in the block *"Approximate Model"* of Figure 5 happens in the following manner. Consider Equation (8) and

1. Fill it in with the parameters of the *available approximate model* and the *actually measured (noise-filtered) values* of the variables $q_1^s$, $q_2^s$, $q_3^s$, and $\dot{q}_1^s$, $\dot{q}_2^s$, $\dot{q}_3^s$;
2. Substitute $Q_3 = 0$ and $\ddot{q}_3^{des}$ into Equation (8d) and calculate $\ddot{q}_2^{des}$;
3. Substitute $\ddot{q}_2^{des}$ and $\ddot{q}_3^{des}$ into Equation (8c) and calculate $Q_2^{appr}$;
4. Substitute $\ddot{q}_1^{des}$ into Equation (8b) and calculate $Q_1^{appr}$.

The lower *"Approximate Model"* block works in similar manner.

For the simulations the following data given in Table 2 were used:

**Table 2.** Controller and simulation data for the robot arm.

| Parameter | Measurement Unit | Value |
|---|---|---|
| Digital time resolution: $\delta t$ | [s] | $10^{-3}$ |
| Noise filtering parameter $\lambda_s$ | $\left[ \mathrm{s}^{-1} \right]$ | $10^3$ |
| Adaptive interpolation parameter $\lambda_a$ | [nondimensional] | 0.9 |
| Trajectory tracking parameter $\Lambda_1$ | $\left[ \mathrm{s}^{-1} \right]$ | 4.0 and 8.0 |
| Trajectory tracking parameter $\Lambda_3$ | $\left[ \mathrm{s}^{-1} \right]$ | 4.0 and 8.0 |
| Norm of augmented vectors $R_a$ | $[\mathrm{N} \cdot \mathrm{m}]$ | $10^4$ |
| $\sigma$ of Gaussian noise | [rad] | 0 and $10^{-5}$ |

### 3.2.1. Simulations without Measurement Noise

The initial transient part during which the initially zero $\dot{q}_1$ and $q_3$ approach their nominal values as well as the fluctuations with which they are kept near the prescribed nominal value depends on the parameters $\Lambda_1$ and $\Lambda_3$. These phases can be well identified in Figure 6. The common norm of the augmented vectors was $R_a = 10^4 \left[ \mathrm{N} \cdot \mathrm{m} \right]$, and the interpolation factor of the adaptivity was $\lambda_a = 0.9$. The time resolution of the simulations (i.e., assumed cycle time of the digital controller) was $\delta t = 10^{-3}$ s, and the simplest Euler integration was applied.

Figure 7 reveals the control forces, the motion of the controlled axle $q_2$, and the angle of the adaptive deformation $\varphi$. Evidently, considerable adaptive deformation was necessary to compensate for the effects of the modeling errors.

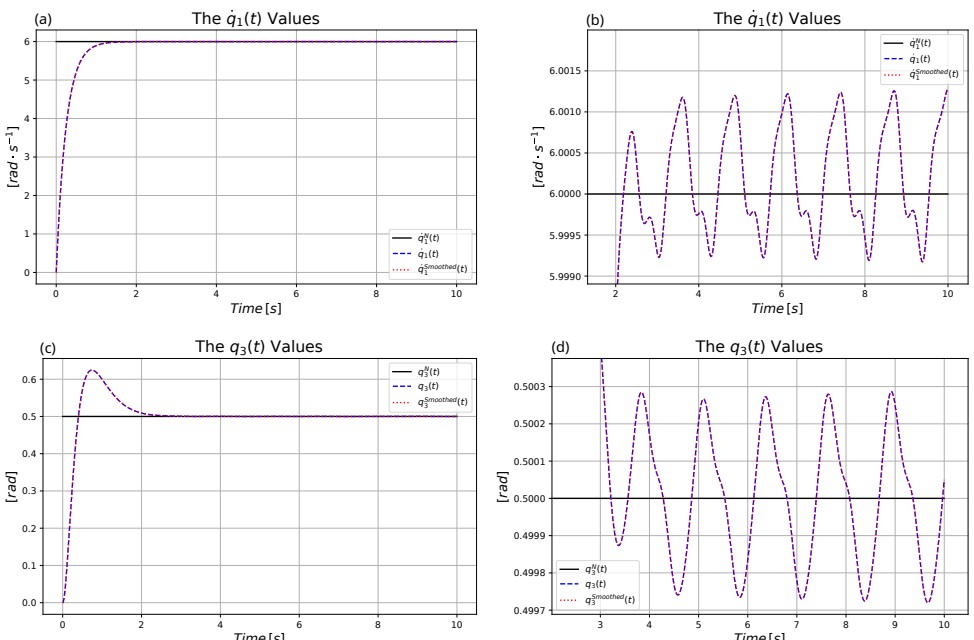

**Figure 6.** The nominal and realized motion for $\Lambda_1 = \Lambda_3 = 4.0\,\text{s}^{-1}$: (**a**) Settling $\dot{q}_1(t)$. (**b**) Small fluctuations in $\dot{q}_1(t)$. (**c**) Settling $q_3(t)$. (**d**) Small fluctuations in $q_3(t)$.

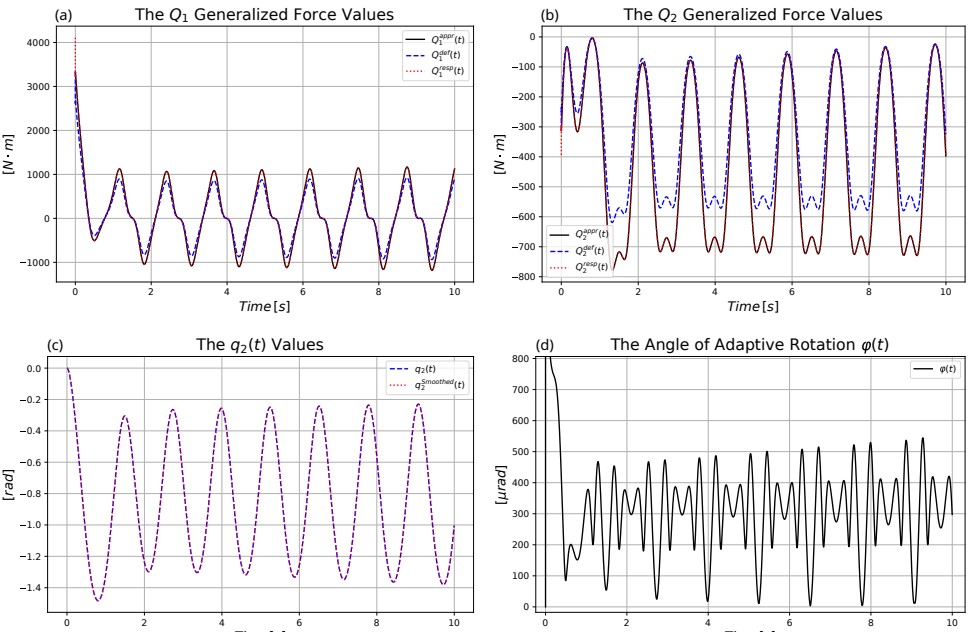

**Figure 7.** Results for $\Lambda_1 = \Lambda_3 = 4.0\,\text{s}^{-1}$: (**a**) The control force $Q_1(t)$. (**b**) The control force $Q_2(t)$. (**c**) The motion of axle $q_2(t)$. (**d**) Angle of the adaptive abstract rotation.

The counterparts of the above figures for $\Lambda_1 = \Lambda_3 = 8.0\,\text{s}^{-1}$ are Figures 8 and 9. They well exemplify the role of the parameters $\Lambda_1$ and $\Lambda_3$.

In these simulations, adaptivity had a key role; without adaptive deformation the controller diverged.

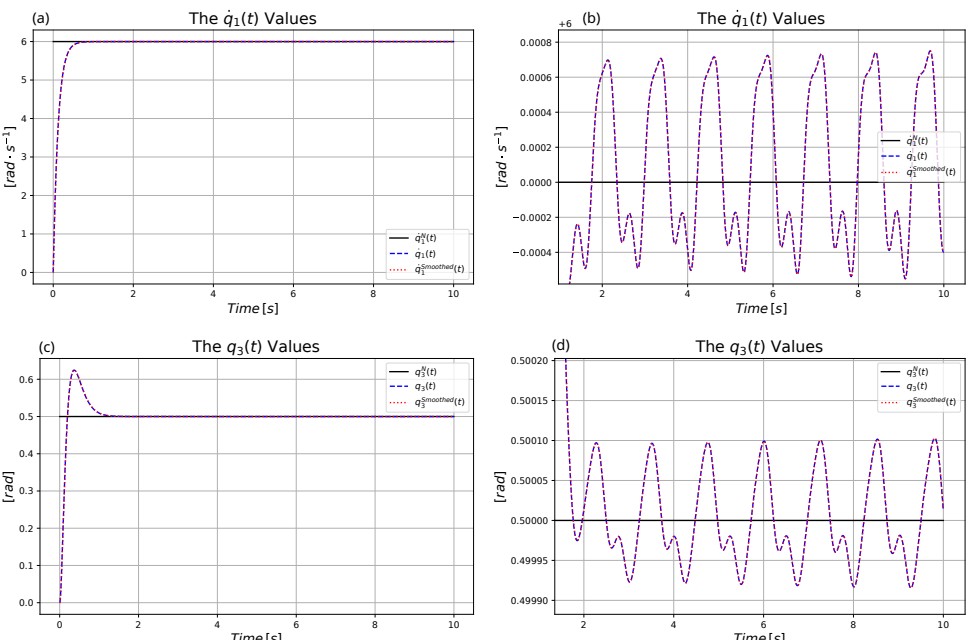

**Figure 8.** The nominal and realized motion for $\Lambda_1 = \Lambda_3 = 8.0\,\mathrm{s}^{-1}$: (**a**) Settling $\dot{q}_1(t)$. (**b**) Small fluctuations in $\dot{q}_1(t)$. (**c**) Settling $q_3(t)$. (**d**) Small fluctuations in $q_3(t)$.

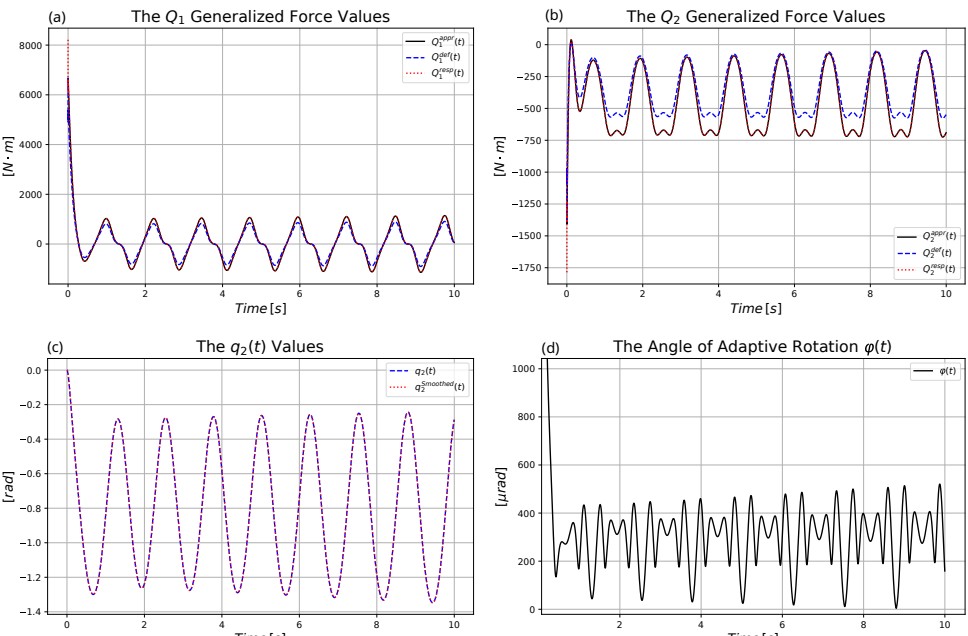

**Figure 9.** Results for $\Lambda_1 = \Lambda_3 = 8.0\,\mathrm{s}^{-1}$: (**a**) The control force $Q_1(t)$. (**b**) The control force $Q_2(t)$. (**c**) The motion of axle $q_2(t)$. (**d**) Angle of the adaptive abstract rotation.

### 3.2.2. Simulations with Measurement Noise

In Figures 10 and 11 the same task is considered as in Section 3.2.1, but in the measurement a Gaussian noise of zero mean and $\sigma = 10^{-5}$ rad standard deviation was assumed for each axle.

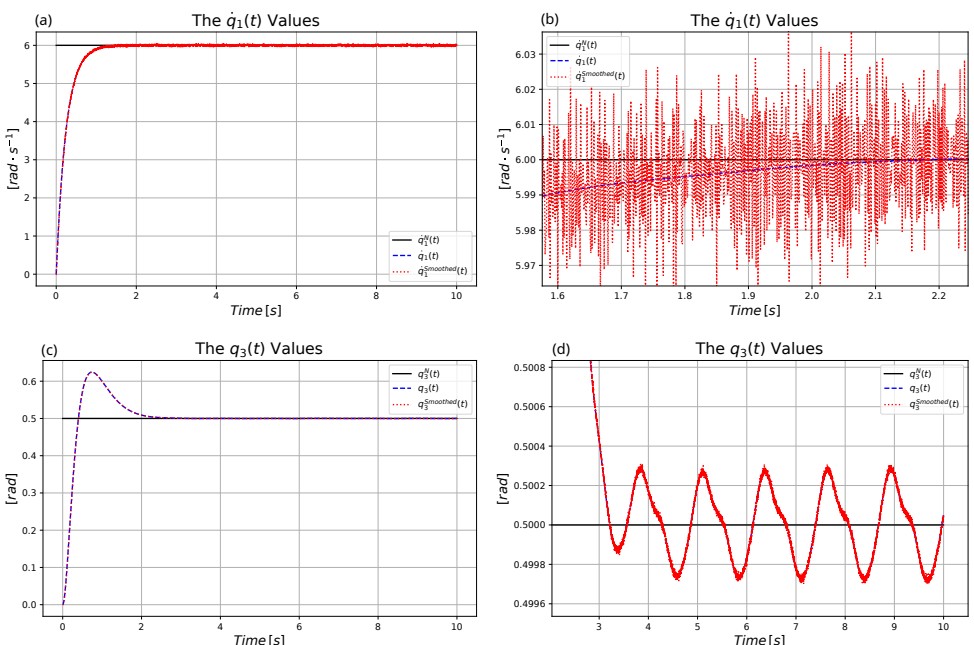

**Figure 10.** The nominal and realized motion under measurement noises: (**a**) Settling $\dot{q}_1(t)$. (**b**) Small fluctuations in $\dot{q}_1(t)$. (**c**) Settling $q_3(t)$. (**d**) Small fluctuations in $q_3(t)$.

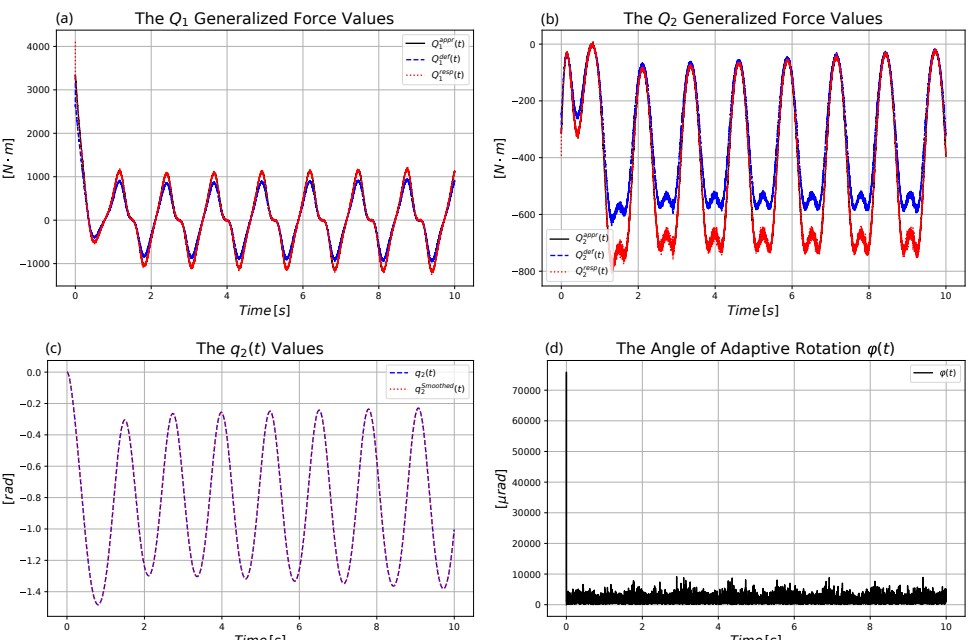

**Figure 11.** (**a**) The control force $Q_1(t)$. (**b**) The control force $Q_2(t)$. (**c**) The motion of axle $q_2(t)$. (**d**) Angle of the adaptive abstract rotation.

It can be concluded that the method was able to tolerate such an order of magnitude noise.

### 3.3. Control of Coupled Non-Linear Springs Increased to Relative Order 3

The system to be controlled is outlined in Figure 12.

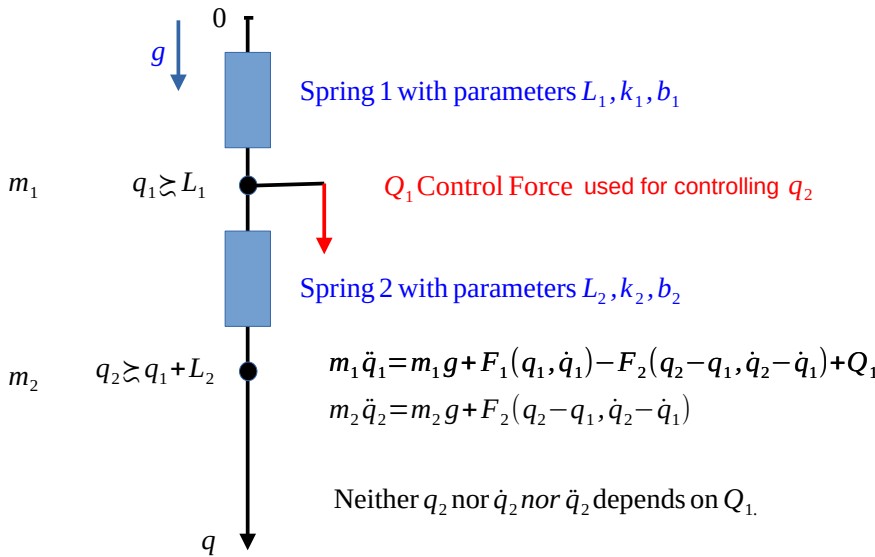

**Figure 12.** The kinematic structure of the coupled springs.

It is assumed that the springs have strongly non-linear model in the sense that they remain "soft" for pulling but strongly harden for compression; furthermore, they have considerable viscous friction in their motion as in Equation (11)

$$F(x, \dot{x}) \overset{def}{=} \begin{cases} -k(x - L) - b\dot{x} \text{ if } x \geq L \ , \\ -kL \ln\left(\frac{x}{L}\right) - b\dot{x} \text{ if } 0 < x < L \ . \end{cases} \tag{11}$$

This spring strongly hardens as its length, $x$, approaches 0. Later, we need the partial derivatives of this function, which are denoted as

$$F_x(x) \overset{def}{=} \frac{\partial F}{\partial x} = \begin{cases} -k \text{ if } x \geq L \ , \\ -\frac{kL}{x} \text{ if } 0 < x < L \end{cases}$$
$$F_y \overset{def}{=} \frac{\partial F}{\partial \dot{x}} = -b = \text{const.} \tag{12}$$

Evidently, due to the viscous friction of Spring 2, $m_2\dddot{q}_2$ will contain $\ddot{q}_1$ that is connected to $Q_1$ by the first equation:

$$m_2\dddot{q}_2 = F_{2x}(\dot{q}_2 - \dot{q}_1) + F_{2y} \cdot (\ddot{q}_2 - \ddot{q}_1) \ , \tag{13}$$

into which $Q_1$ can be substituted from the first equation of motion. Therefore, as a starting point of this task, the following three equations can be considered:

$$\ddot{q}_1 = g + \frac{1}{m_1}F_1(q_1, \dot{q}_1) - \frac{1}{m_1}F_2(q_2 - q_1, \dot{q}_2 - \dot{q}_1) + \frac{1}{m_1}Q_1 \ , \tag{14a}$$

$$\ddot{q}_2 = g + \frac{1}{m_2}F_2(q_2 - q_1, \dot{q}_2 - \dot{q}_1) \ , \tag{14b}$$

$$\dddot{q}_2 = \frac{1}{m_2}F_{2x}(q_2 - q_1) \cdot (\dot{q}_2 - \dot{q}_1) + \frac{1}{m_2}F_{2y} \cdot \left(\ddot{q}_2 - g - \frac{1}{m_1}F_1(q_1, \dot{q}_1) + \frac{1}{m_1}F_2(q_2 - q_1, \dot{q}_2 - \dot{q}_1) - \frac{1}{m_1}Q_1\right) \ . \tag{14c}$$

From this, it follows that the *relative order of our control task is 3*, because the value of $\dddot{q}_2(t)$ can be instantaneously controlled by the control force $Q_1(t)$. However, this task is different to a *normal order 3* control, since in the normal case $\dddot{q}_2(t)$ **can be set independently of** $q_2(t)$, $\dot{q}_2(t)$, and $\ddot{q}_2(t)$. In our case $\dddot{q}_2(t)$ **cannot be set independently of these values**: the system's dynamic model determines $\dddot{q}_2(t)$ by its lower order derivatives, and by $Q_1$. **This easily may lead to inconsistent kinematic requirements if the model parameters**

**are not exactly known and, e.g., the stiffness of the non-linear spring in our case can be divergent function of the spring's length.** Because of this, the controller's structure outlined in Figure 13 is suggested to tackle this problem.

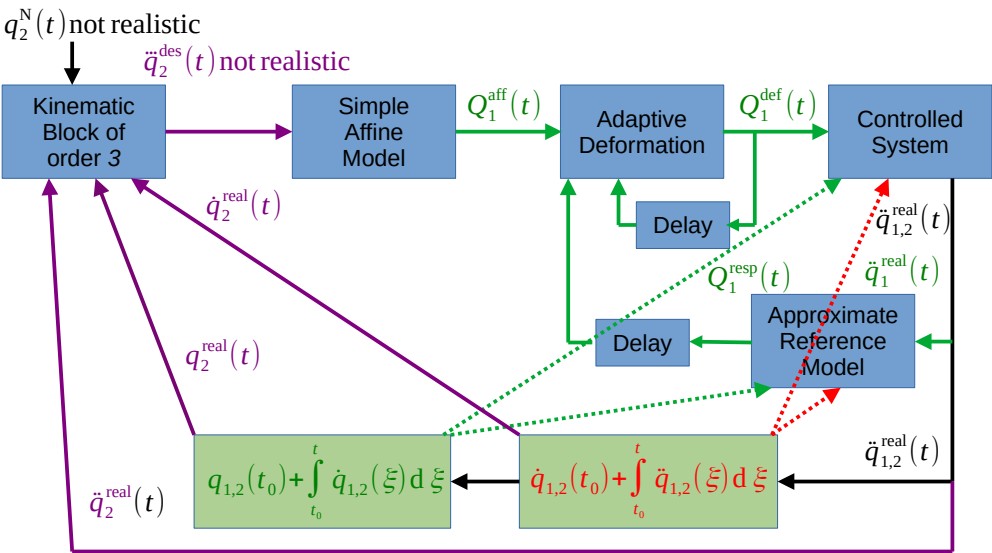

**Figure 13.** The control structure of the underactuated coupled springs.

In Figure 13, instead of the complicated inverse of Equation (14c) the simple affine model is used for the calculation of the reference force $Q_1^{aff}(t)$ in the box "*Simple Affine Model*" as

$$Q_1^{aff}(t) \approx \mathfrak{A}\ddot{q}_2^{des}(t) + \mathfrak{B} \tag{15}$$

with the parameters $\mathfrak{A}$ and $\mathfrak{B}$, the effect of which, together with that of the kinematic inconsistencies, can be compensated by the adaptivity of the controller. Since $Q_1^{def}(t)$, i.e., *the actually applied control force*, immediately determines $\ddot{q}_1(t)$ according to the exact dynamic model parameters, the adaptation can be closed in the green loop related to $Q_1^{resp}$ and $\ddot{q}_1$ according to Equation (14a) using the available approximate dynamic parameters in the box "*Approximate Reference Model*". This will produce a *realizable motion* for $q_1(t)$ that is influenced by the not completely realistic $\ddot{q}_2^{des}(t)$ requirement. As a result the not realistic prescription will be well approximated by a realistic one. In the simulations, Euler integration happens according to Equations (14a) and (14b) using the exact dynamic model parameters.

For the kinematic prescription to vanish the tracking error, assume that we have positive constants $\lambda_k > \Lambda$, and to obtain $\dddot{q}_2(t)$ prescribe in the box "*Kinematic Block of order 3*" Equation (16) as

$$\left(\lambda_k + \frac{\mathrm{d}}{\mathrm{d}t}\right)^3 \left(\Lambda + \frac{\mathrm{d}}{\mathrm{d}t}\right) e_{int} \equiv 0 \tag{16}$$

Since the solution of $\left(\lambda_k + \frac{\mathrm{d}}{\mathrm{d}t}\right) g(t) = 0$ is $g(t) = \exp(-\lambda_k(t - t_0))g(t_0) \to 0$ as $t \to \infty$, if this strategy is realized due to adaptivity, after a while the system arrives at $\left(\Lambda + \frac{\mathrm{d}}{\mathrm{d}t}\right) e_{int} \equiv 0$ that yields a monotonic reduction in the integrated error.

For this system of relative order 3 the following order 4 noise filtering was applied (Equation (17)):

$$\left(\lambda_s + \frac{\mathrm{d}}{\mathrm{d}t}\right)^4 q_i^s(t) = \lambda_s^4 q_i^o(t) \tag{17}$$

with the initial conditions $q_i^s(t_0) = q_i(t_0) = 0$, $\dot{q}_i^s(t_0) = \dot{q}_i(t_0) = 0$, and $\ddot{q}_i^s(t_0) = \ddot{q}_i(t_0) = 0$, $\dddot{q}_i^s(t_0) = \dddot{q}_i(t_0) = 0$ with $\lambda_s = 10^3 \, [\text{s}^{-1}]$. For modeling measurement noises Gaussian noise of zero mean and $\sigma = 10^{-5}$ m was assumed.

In the simulations, the model data given in Table 3 were used.

**Table 3.** The kinematic and dynamic parameters of the coupled springs.

| Parameter | Measurement Unit | Exact Value | Approx. Value |
|---|---|---|---|
| Length of spring 1: $L_1$ | [m] | 1.0 | 0.8 |
| Length of spring 2: $L_2$ | [m] | 1.5 | 1.3 |
| Mass of point 1: $m_1$ | [kg] | 1.5 | 2.0 |
| Mass of point 2: $m_2$ | [kg] | 2.0 | 2.5 |
| Spring stiffness 1: $k_1$ | $[\text{N} \cdot \text{m}^{-1}]$ | 100.0 | 90.0 |
| Spring stiffness 2: $k_2$ | $[\text{N} \cdot \text{m}^{-1}]$ | 200.0 | 190.0 |
| Gravitational acceleration: $g$ | $[\text{m} \cdot \text{s}^{-2}]$ | 9.81 | 9.81 |
| Viscous damping 1: $b_1$ | $[\text{N} \cdot \text{s} \cdot \text{m}^{-1}]$ | 2.5 | 2.0 |
| Viscous damping 2: $b_2$ | $[\text{N} \cdot \text{s} \cdot \text{m}^{-1}]$ | 3.0 | 2.5 |
| Affine parameter 1: $\mathfrak{A}$ | $[\text{N} \cdot \text{s}^3 \cdot \text{m}^{-1}]$ | Not applicable | various |
| Affine parameter 2: $\mathfrak{B}$ | [N] | Not applicable | $-15.0$ |

For setting the parameters it was assumed that the gravitational acceleration is known, but the other model data should have considerable approximation errors as underestimated "zero force lengths" for the springs, overestimated masses, underestimated spring stiffness values, and viscous friction terms.

For the spring system, the numerical data given in Table 4 were used with a deformed and shifted sinusoidal nominal trajectory determined by Equation (18) as

$$q_2^N(t) = \tanh\left(\frac{t}{T}\right) A_o \tanh(A_i \sin \Omega t) + S \tag{18}$$

In this manner, more complex motions can be considered than the simple sinusoidal ones. Furthermore, the "rule" that it is not expedient to apply an initial shock to a nonlinear system is taken into consideration too. (Since linear time-invariant systems have only additional transients, in their case arbitrary initial shocks, such as abrupt jumps in the nominal trajectory to be tracked, can be applied without significant consequences). For the simulations the data given in Table 4 were used.

**Table 4.** Controller and simulation data for the coupled springs.

| Parameter | Measurement Unit | Value |
|---|---|---|
| Digital time resolution: $\delta t$ | [s] | $10^{-3}$ |
| Noise filtering parameter $\lambda_s$ | $[\text{s}^{-1}]$ | $10^3$ |
| Adaptive interp. param. $\lambda_a$ | [nondimensional] | 0.9, 0.05 affine, complex |
| Trajectory tracking parameter $\Lambda$ | $[\text{s}^{-1}]$ | 4.0 |
| Trajectory tracking parameter $\lambda_k$ | $[\text{s}^{-1}]$ | 12.0 |
| Norm of augmented vectors $R_a$ | [N] | $10^{-6}$ |
| $\sigma$ of Gaussian noise | [m] | 0 and $10^{-6}$ |
| Nominal trajectory rise time $T$ | [s] | 3.0 |
| Nominal trajectory amplitude $A_o$ | [m] | $0.2 \cdot (L_1 + L_2)$ (exact) |
| Nominal trajectory deformation factor $A_i$ | [nondimensional] | 2.0 |
| Nominal trajectory circular frequency $\Omega$ | $[\text{s}^{-1}]$ | 2.0 |
| Nominal trajectory shift $S$ | [m] | $(L_1 + L_2)$ (exact) |

### 3.3.1. Simulations for the Affine Model without Measurement Noise

At first, the role of the adaptive deformation in using the simple affine model is illustrated. In Figure 14, it can be observed that without the adaptation a long-lasting fluctuation in the control force and in the motion of coordinate $q_1$ was generated.

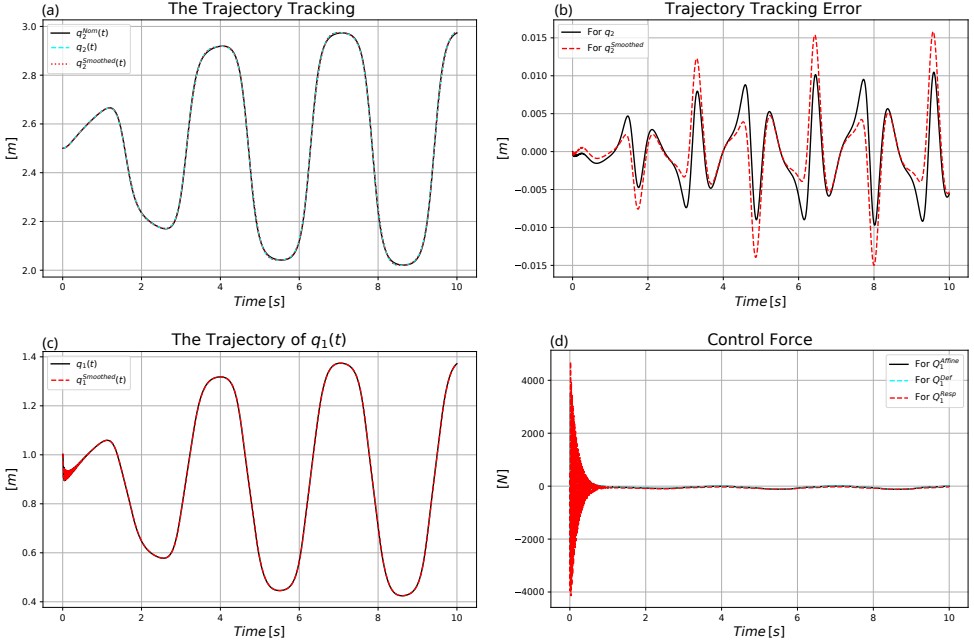

**Figure 14.** Control of the affine model with $\mathfrak{A} = 7.0 \left[ \text{N} \cdot \text{s}^3 \cdot \text{m}^{-1} \right]$ without adaptivity: (**a**) Trajectory tracking. (**b**) Trajectory tracking error. (**c**) The motion of mass point 1. (**d**) The control force.

On the basis of Figures 14–17 the following observations can be performed:

1.  While the tracking error remained in the same range, *the illusion of the MRAC control, i.e., that on the basis of the kinematic prescriptions the dynamics of the affine model was controlled* was well provided by the solution. The affine model was used by the external kinematic loop for the computation of the necessary control force, and precise trajectory tracking was achieved. The affine model's force need was approximately in the range $[-120, -30]$ N following the transient phase, while the actual control force varied within the range $[-120, 5]$ N.
2.  Due to the adaptivity the duration of the initial "transient swinging phase" of the control force was considerably reduced.
3.  The duration of the initial transient swinging in the position of mass point 1 was considerably reduced due to the adaptivity.
4.  Furthermore, the application of adaptivity considerably reduced the amplitude of the control force in the initial transient phase of the motion from the range $[-4000, 4500]$ N to $[-3000, 2500]$ N.

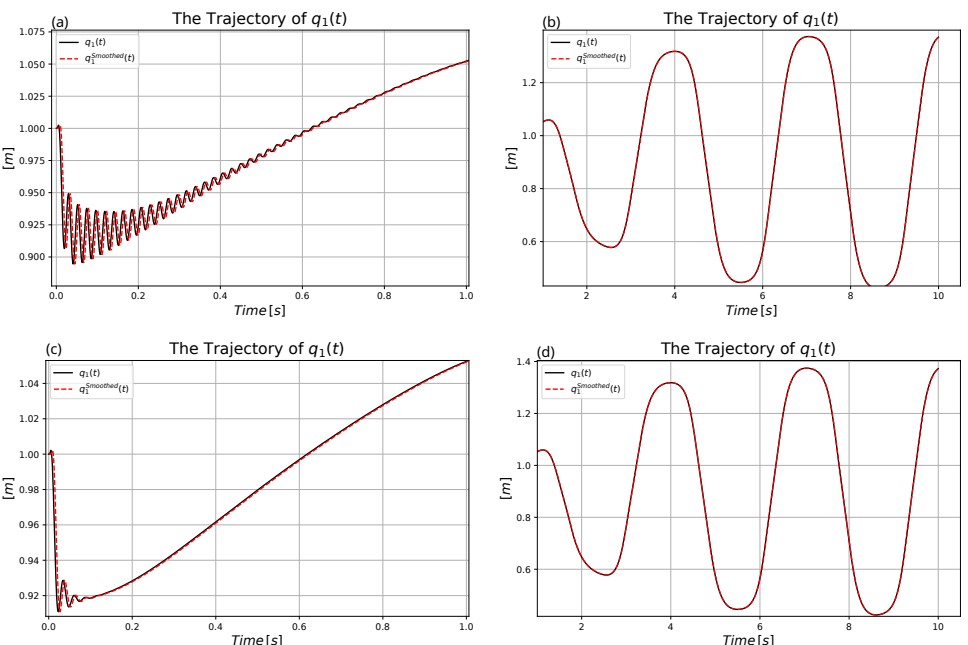

**Figure 15.** Comparison of the motion of $q_1$ for the adaptive and the non-adaptive control for $\mathfrak{A} = 7.0 \left[ \text{N} \cdot \text{s}^3 \cdot \text{m}^{-1} \right]$. (**a**) Non-adaptive motion of $q_1$ in the first second. (**b**) Non-adaptive motion of $q_1$ in the rest of the trajectory. (**c**) Adaptive motion of $q_1$ in the first second. (**d**) Adaptive motion of $q_1$ in the rest of the trajectory.

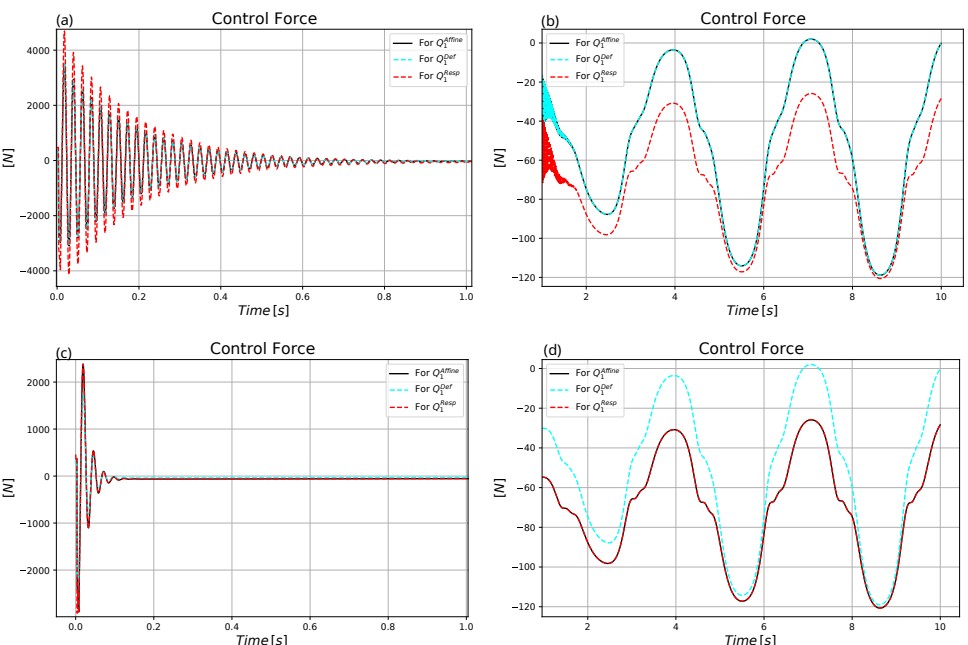

**Figure 16.** Comparison of the control force $Q_1$ for the adaptive and the non-adaptive control for $\mathfrak{A} = 7.0 \left[ \text{N} \cdot \text{s}^3 \cdot \text{m}^{-1} \right]$. (**a**) Non-adaptive motion in the first second. (**b**) Non-adaptive motion in the rest of the trajectory. (**c**) Adaptive motion in the first second. (**d**) Adaptive motion in the rest of the trajectory.

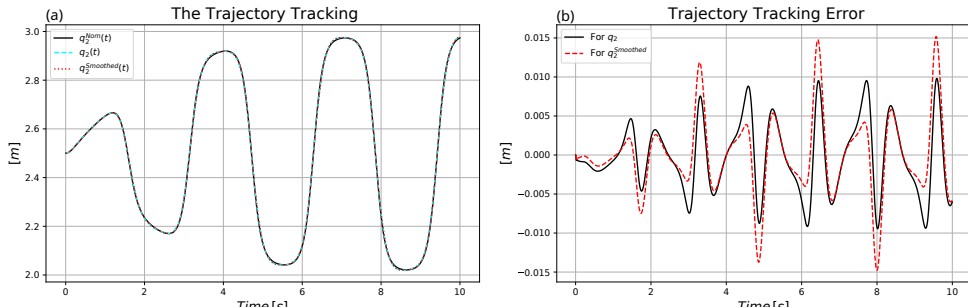

**Figure 17.** Control of the affine model with $\mathfrak{A} = 7.0 \left[ \text{N} \cdot \text{s}^3 \cdot \text{m}^{-1} \right]$ and adaptivity: (**a**) Trajectory tracking. (**b**) Trajectory tracking error.

The next figures (Figures 18–20) reveal the results for the $\mathfrak{A} = 5.0 \left[ \text{N} \cdot \text{s}^3 \cdot \text{m}^{-1} \right]$ affine parameter. The same observations can be made in the case of these results as in the case of the set belonging to $\mathfrak{A} = 7.0 \left[ \text{N} \cdot \text{s}^3 \cdot \text{m}^{-1} \right]$.

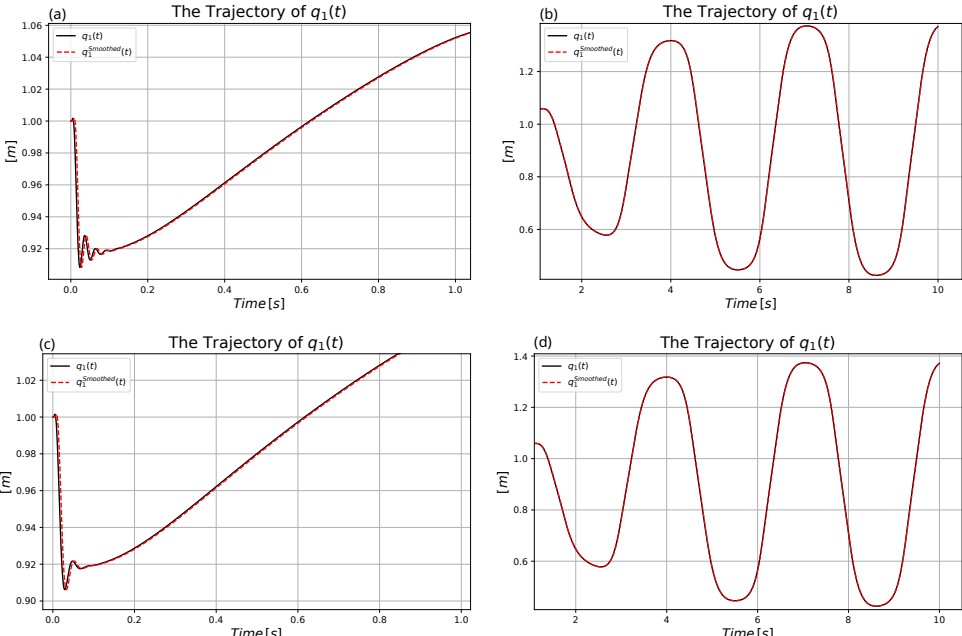

**Figure 18.** Comparison of the motion of $q_1$ for the adaptive and the non-adaptive control for $\mathfrak{A} = 5.0 \left[ \text{N} \cdot \text{s}^3 \cdot \text{m}^{-1} \right]$. (**a**) Non-adaptive motion of $q_1$ in the first second. (**b**) Non-adaptive motion of $q_1$ in the rest of the trajectory. (**c**) Adaptive motion of $q_1$ in the first second. (**d**) Adaptive motion of $q_1$ in the rest of the trajectory.

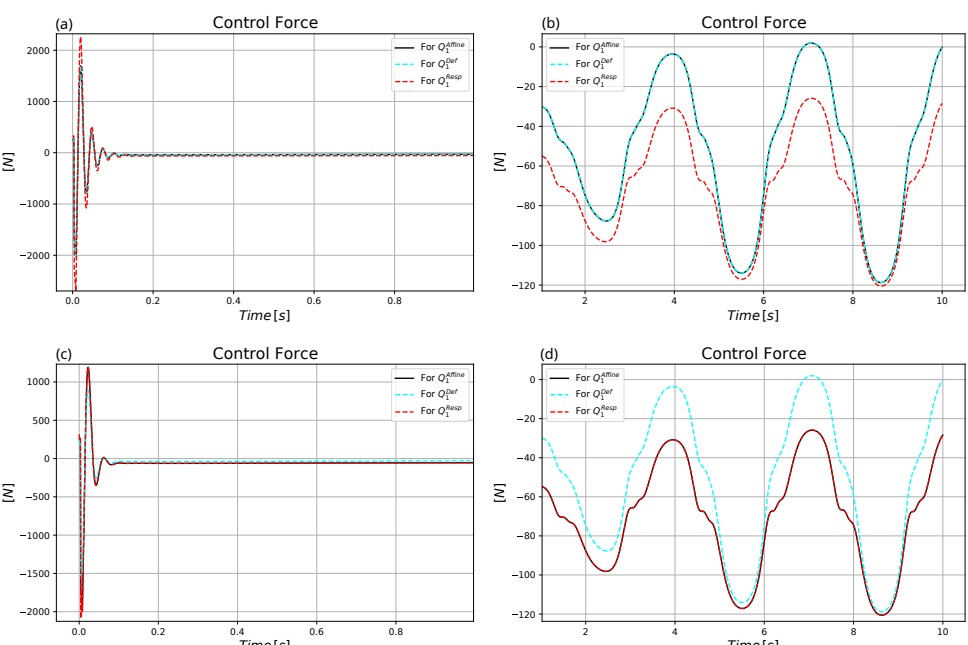

**Figure 19.** Comparison of the control force $Q_1$ for the adaptive and the non-adaptive control for $\mathfrak{A} = 5.0 \left[ \text{N} \cdot \text{s}^3 \cdot \text{m}^{-1} \right]$. (**a**) Non-adaptive motion in the first second. (**b**) Non-adaptive motion in the rest of the trajectory. (**c**) Adaptive motion in the first second. (**d**) Adaptive motion in the rest of the trajectory.

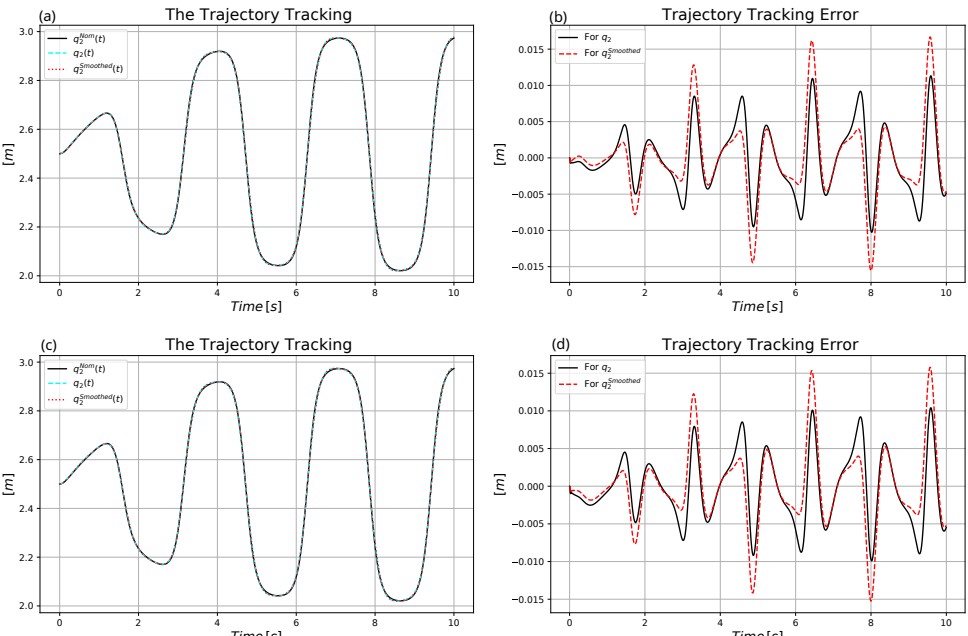

**Figure 20.** Control of the affine model with $\mathfrak{A} = 5.0 \left[ \text{N} \cdot \text{s}^3 \cdot \text{m}^{-1} \right]$ (**a**) Non-adaptive trajectory tracking. (**b**) Non-adaptive trajectory tracking error. (**c**) Adaptive trajectory tracking. (**d**) Adaptive trajectory tracking error.

In the following section, the effects of an extremely low value, $\mathfrak{A} = 0.4 \, \text{s}^3 \cdot \text{m}^{-1}$ are investigated. Figures 21–23 reveal that in this case, when very drastic inconsistency is present between the affine model and the realistic one, the adaptive controller produces more hectic variation of coordinate $q_1(t)$ and in the control force. However, the MRAC illusion is well maintained, since the affine and the response forces are very close to each

other, and considerably differ from the actual control (i.e., the deformed) forces. In this case the precision of the trajectory tracking is a little bit improved.

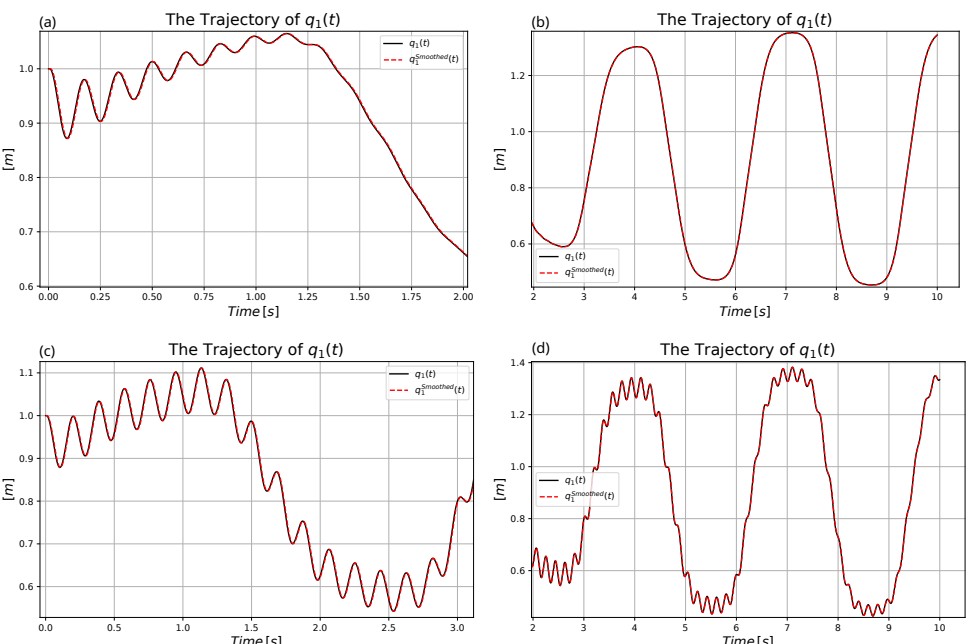

**Figure 21.** Comparison of the motion of $q_1$ for the adaptive and the non-adaptive control for $\mathfrak{A} = 0.4 \left[ \text{N} \cdot \text{s}^3 \cdot \text{m}^{-1} \right]$ (**a**) Non-adaptive motion of $q_1$ in the first two seconds. (**b**) Non-adaptive motion of $q_1$ in the rest of the trajectory. (**c**) Adaptive motion of $q_1$ in the first three seconds. (**d**) Adaptive motion of $q_1$ in the rest of the trajectory.

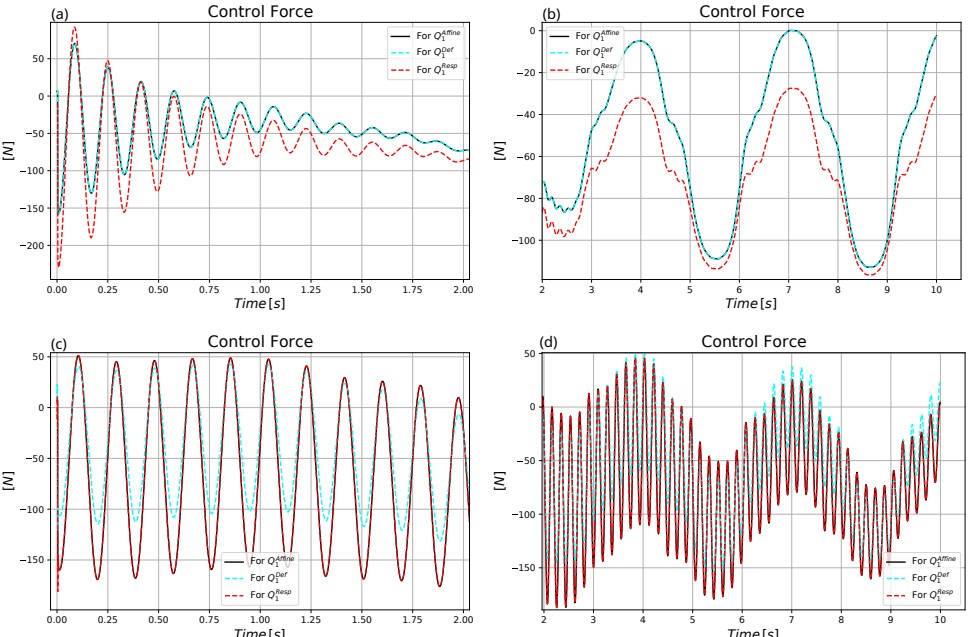

**Figure 22.** Comparison of the control force $Q_1$ for the adaptive and the non-adaptive control for $\mathfrak{A} = 0.4 \left[ \text{N} \cdot \text{s}^3 \cdot \text{m}^{-1} \right]$ (**a**) Non-adaptive motion in the first two seconds. (**b**) Non-adaptive motion in the rest of the trajectory. (**c**) Adaptive motion in the first two seconds. (**d**) Adaptive motion in the rest of the trajectory.

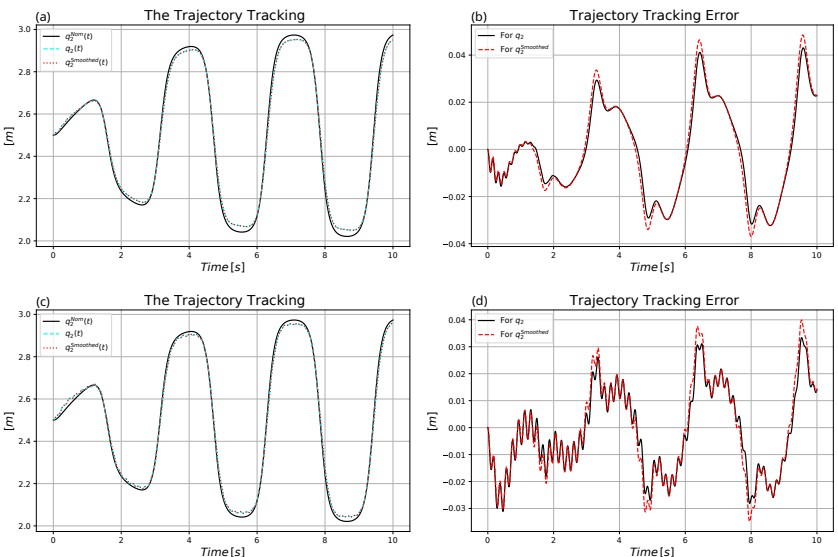

**Figure 23.** Control of the affine model with $\mathfrak{A} = 0.4 \left[ \mathrm{N \cdot s^3 \cdot m^{-1}} \right]$ (**a**) Non-adaptive trajectory tracking. (**b**) Non-adaptive trajectory tracking error. (**c**) Adaptive trajectory tracking. (**d**) Adaptive trajectory tracking error.

As it can be expected from the dynamic model, the necessary control forces mainly depend on the amplitude of the nominal motion that directly concerns the spring dilatation/compression values, and the time-derivatives of the coordinates that generate the friction forces. The affine parameter $\mathfrak{A}$ mainly determines the duration of the initial oscillating phase. It can be noted, too, that in harmony with the expectation for the "*approximately differentially direction keeping*" response function, for $\mathfrak{A} < 0$ and too small $\mathfrak{A} > 0$ the adaptive controller became divergent.

### 3.3.2. Simulations for the Affine Model with Measurement Noise

In this case, non-adaptive and adaptive simulations were made for $\mathfrak{A} = 7.0 \left[ \mathrm{N \cdot s^3 \cdot m^{-1}} \right]$. Figures 24 and 25 reveal the chaotic fluctuation in the control force that does not completely destroy adaptivity.

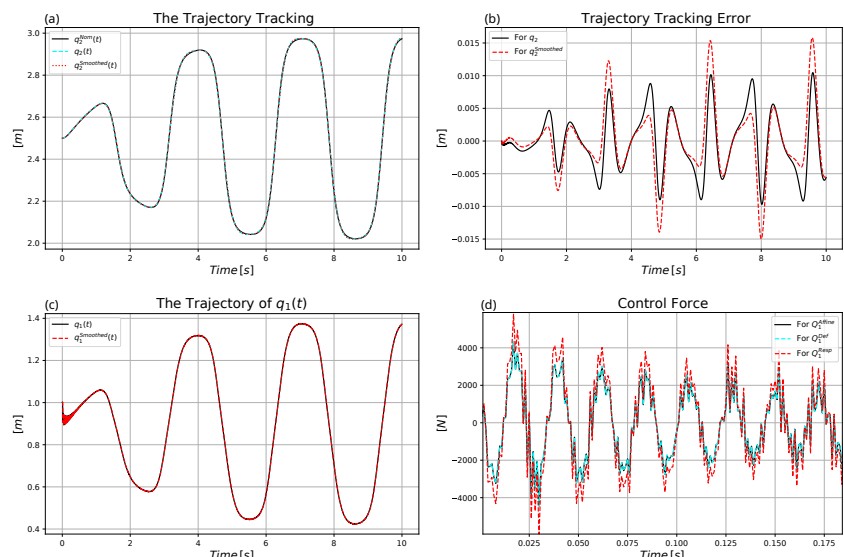

**Figure 24.** Control of the affine model with $\mathfrak{A} = 7.0 \left[ \mathrm{N \cdot s^3 \cdot m^{-1}} \right]$ without adaptivity under measurement noises: (**a**) Trajectory tracking. (**b**) Trajectory tracking error. (**c**) The motion of mass point 1. (**d**) The control force (zoomed in excerpt).

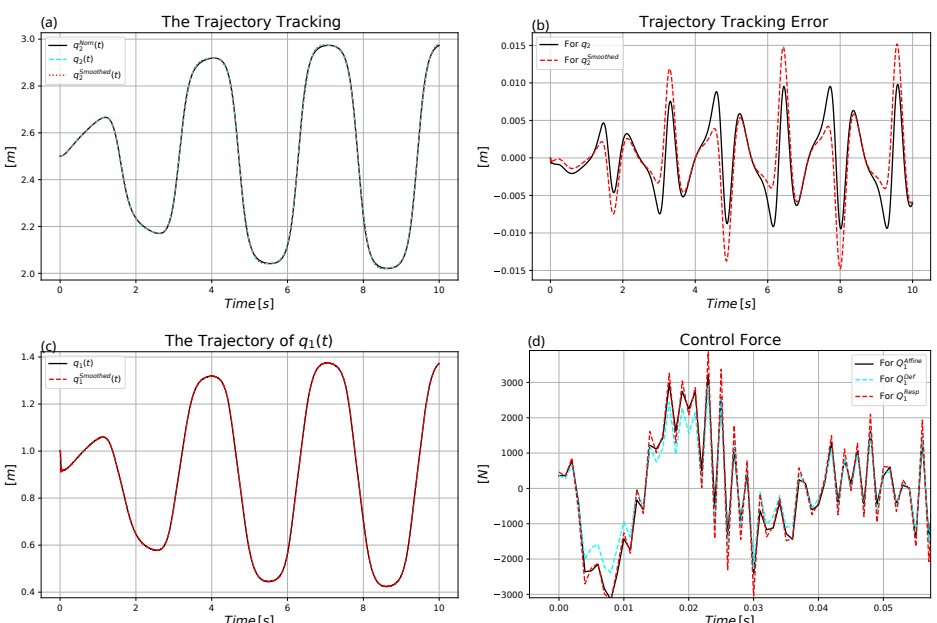

**Figure 25.** Control of the affine model with $\mathfrak{A} = 7.0 \left[\mathrm{N \cdot s^3 \cdot m^{-1}}\right]$ with adaptivity under measurement noises: (**a**) Trajectory tracking. (**b**) Trajectory tracking error. (**c**) The motion of mass point 1. (**d**) The control force (zoomed in excerpt).

### 3.3.3. Simulations for the Complex Order 3 Model without Measurement Noise

For comparison the original version of the FPI-based MRAC controller for fully actuated systems (Figure 3) has been modified for the order 3 underactuated version in Figure 26. In this case, the inverse of Equation (14c), i.e., Equation (19) is used in the boxes "*Complex Reference Model*" with the available approximate model parameters.

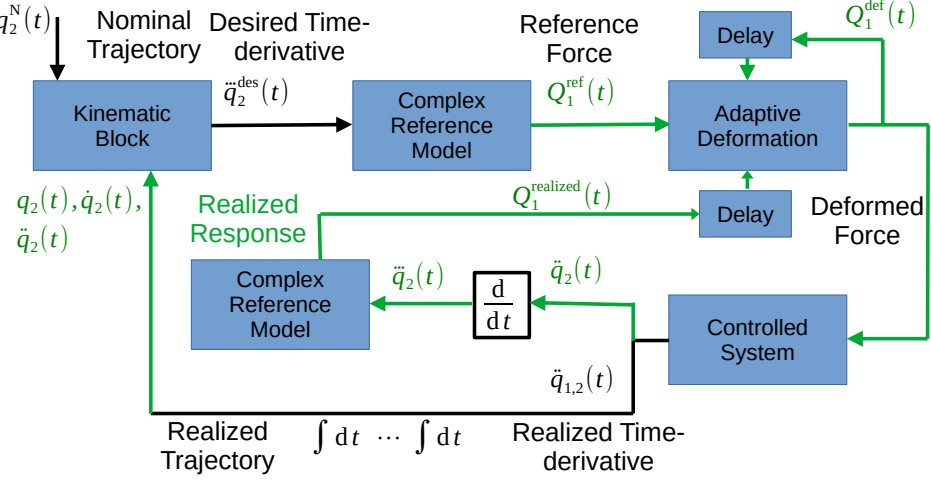

**Figure 26.** The control structure of the underactuated coupled springs using the complex order 3 model: the input of the "*Complex Reference Model*" corresponds to the function in Equation (19). For the sake of clarity the figure does not contain each input.

$$Q_1 = \frac{-m_1}{F_{2y}} \left[ m_2 \dddot{q}_2 - F_{2x}(q_2 - q_1)(\dot{q}_2 - \dot{q}_1) - F_{2y} \left\{ \ddot{q}_2 - g - \frac{1}{m_1} F_1(q_1, \dot{q}_1) \right. \right.$$
$$\left. \left. + \frac{1}{m_1} F_2(q_2 - q_1, \dot{q}_2 - \dot{q}_1) \right\} \right] \; . \tag{19}$$

In the simulations for the input of Equation (19) the noise-filtered estimations of $q_1$, $q_2$, $\dot{q}_1$, $\dot{q}_2$, and $\ddot{q}_2$ are used. The Julia language code excerpt of this function is given below

```
function Q_3rdOrdModel(q2_pppDes,q1,q2,q1_p,q2_p,q2_pp)
   local casual
   casual=m2a*q2_pppDes-F2xa(q2-q1)*(q2_p-q1_p)
   casual-=F2ya*(q2_pp-ga-F1a(q1,q1_p)/m1a+F2a(q2-q1,q2_p-q1_p)/m1a)
   return -m1a*casual/F2ya
end
```

in which q1 and q2 stand for $q_1$ and $q_2$, q1_p and q2_p is in the role of $\dot{q}_1$ and $\dot{q}_2$, and q2_pp represents $\ddot{q}_2$, m1a, m2a, and ga denote the approximate parameter values of $m_1$, $m_2$, and $g$, F2xa and F2ya correspond to the functions $F_{2x}$ and $F_{2y}$ in Equation (12) with the approximate model parameters. In a similar manner, the functions F1a and F2a are the counterparts of the functions in Equation (11).

The following conclusions can be made regarding the results of the noise-free simulations:

1.  The tracking precision for the simple affine reference model with $\mathfrak{A} = 7.0 \left[ \text{N} \cdot \text{s}^3 \cdot \text{m}^{-1} \right]$ is in the same range as in the case of the complicated order 3 model (compare Figures 14, 17 and 27).
2.  The actually exerted forces are essentially the same that is determined by the desired motion, observable differences appear only in the "MRAC illusions" provided by the different solutions (compare Figures 16 and 28).
3.  By the use of the affine model the initial transients were successfully reduced (compare Figures 15 and 29).

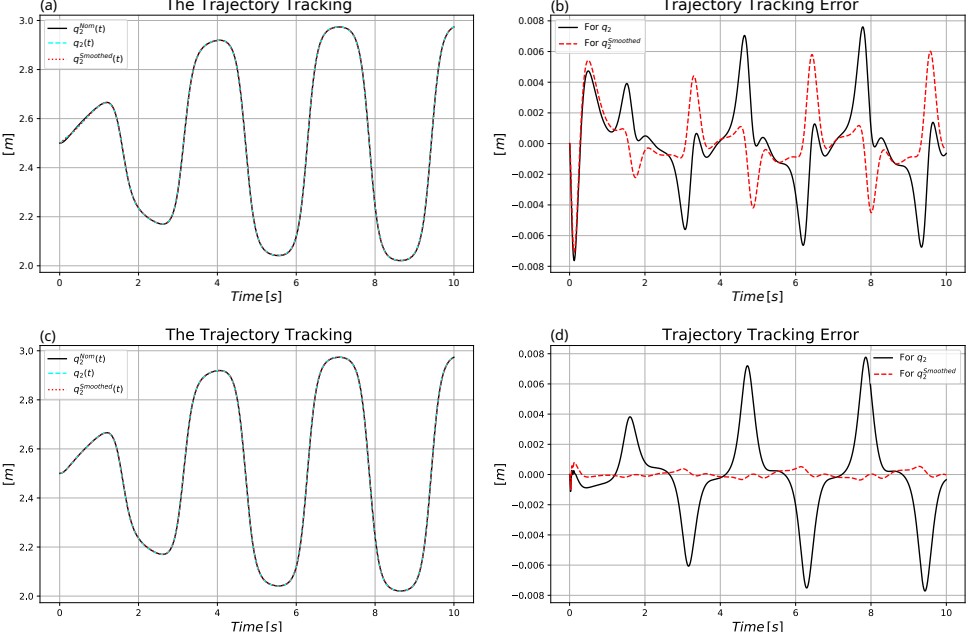

**Figure 27.** Control of the complex model without measurement noises: (**a**) Non-adaptive trajectory tracking. (**b**) Non-adaptive trajectory tracking error. (**c**) Adaptive trajectory tracking. (**d**) Adaptive trajectory tracking error.

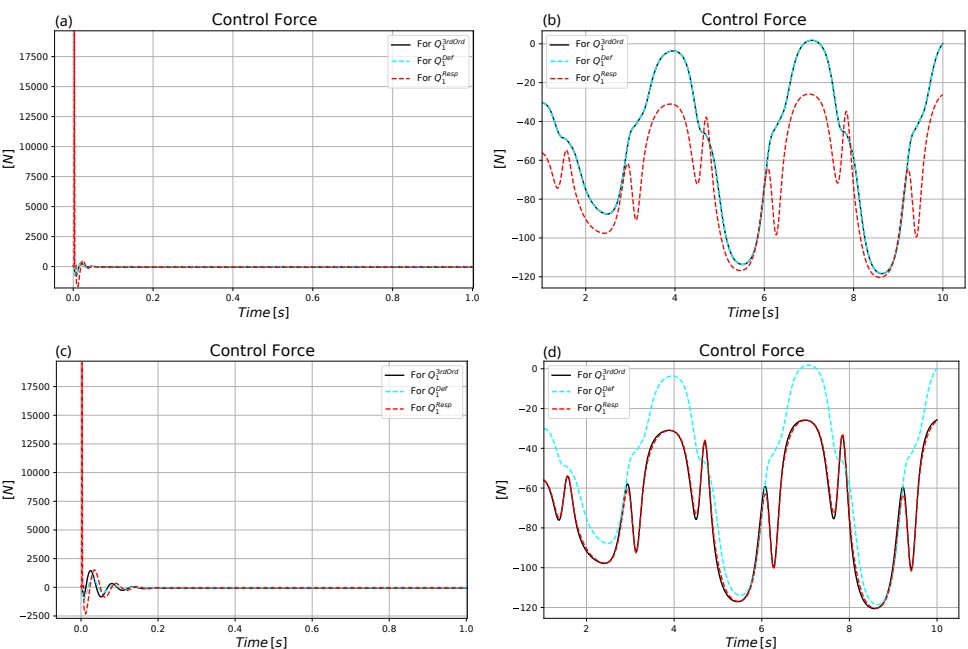

**Figure 28.** Comparison of the control force $Q_1$ for the adaptive and the non-adaptive control for the complex model without measurement noises. (**a**) Non-adaptive motion in the first second. (**b**) Non-adaptive motion in the rest of the trajectory. (**c**) Adaptive motion in the first second. (**d**) Adaptive motion in the rest of the trajectory.

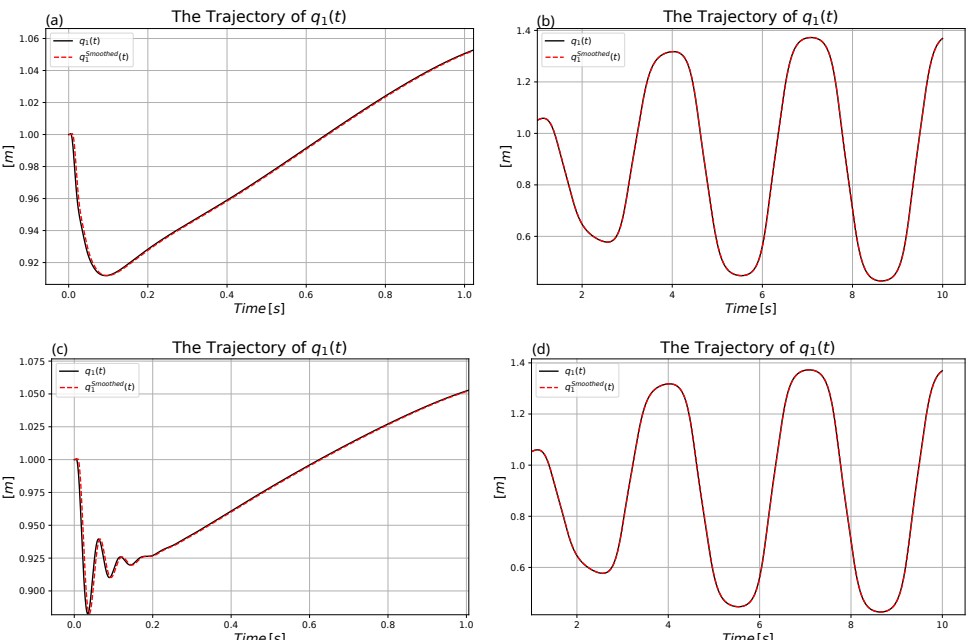

**Figure 29.** Comparison of the motion of $q_1$ for the adaptive and the non-adaptive control for the complex model without measurement noises (**a**) Non-adaptive motion of $q_1$ in the first second. (**b**) Non-adaptive motion of $q_1$ in the rest of the trajectory. (**c**) Adaptive motion of $q_1$ in the first second. (**d**) Adaptive motion of $q_1$ in the rest of the trajectory.

3.3.4. Simulations for the Complex Order 3 Model with Measurement Noise

Certain results with measurement noises are given in Figure 30. The comparison with Figure 25 reveals that the attempt of feeding back high order derivatives and using them in the adaptive iteration makes the method very noise-sensitive. Though the exerted (deformed) control forces are not too high, in the case of the complex model using various

derivatives as the input does not provide some clear "MRAC illusion", since the "response force" has huge noise. However, in the case of the simple affine model with lower order adaptive feedback this function can be identified in the computational results.

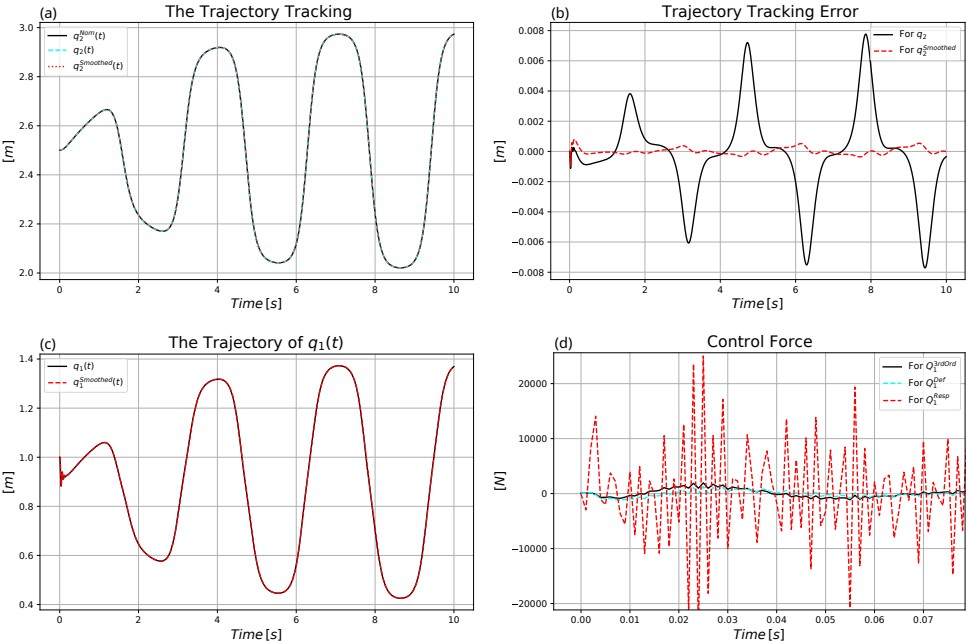

**Figure 30.** Control of the complex model with adaptivity under measurement noises: (**a**) Trajectory tracking. (**b**) Trajectory tracking error. (**c**) The motion of mass point 1. (**d**) The control force (zoomed in excerpt).

### 3.4. Summary of the Innovation

In this paper, the history of the formation of the here suggested control method was outlined step-by-step. It was shown how the FPIAC controller had been transformed into the FPI-based MRAC controller for controlling *fully actuated systems*. The present innovation is the modification of this latter method to control *underactuated systems* by the application of the adaptive deformation algorithm to *only a reduced number of the generalized coordinates of the controlled system*.

Furthermore, the increased relative order task allowed the use of a simple primitive affine model with lower order adaptive feedback instead of the calculation of the complicated model terms on the basis of approximate model parameters. In this manner, the method produced shortened initial swinging and considerably reduced noise sensitivity in comparison with the original higher order approach.

The primitive affine model is so simple that it can be realized in an embedded system. Together with the approximate model parameters used in the computations it provided a fictive "reference system" by which the inconsistencies between the high relative order kinematic design and the dynamic properties of the model were eliminated or evaded.

### 4. Conclusions

This paper systematically investigated the FPIAC method to adaptively control approximately modeled underactuated systems. In the suggested solution, the iteration that yielded the appropriate control signal was moved from the space of the time-derivatives of the generalized coordinates to the space of the generalized forces as in the case of the FPI-based MRAC controllers developed for fully actuated systems. However, the loop of adaptive deformation was applied only for the generalized coordinates that played independent roles in the control of the underactuated systems.

It was shown that when the underactuation causes an increase in the relative order of the control, the kinematic specifications separated from the dynamic model may be incompatible with the physical capabilities of the controlled system. This discrepancy was resolved by the application of further convenient simplification. Instead, making an attempt to estimate the higher order derivative of interest by the use of Lie derivatives and the available approximate system model a *simple affine* model part was introduced. The integrated effects of these approximations were compensated by the adaptive deformation.

Two typical examples were investigated by Julia language-based simulations, when the underactuation was not accompanied by an increase in the relative order of the controller and when the relative order was increased. For the first case, the dynamic model of a 3-degree-of-freedom robot arm was considered with a corrupted drive. For the second case, dynamically coupled strongly non-linear springs were modeled.

Though it can be expected that due to the "not conventional feedback terms", the system must be noise sensitive, it has been shown by the simulations that simple low pass filters can be incorporated into the controller so that it remains convergent under little standard deviation of the noise that appears in the measurement of the generalized coordinates' values.

In addition to potential noise sensitivity, the main drawback of the method is that during one digital control step only one step of the adaptive deformation can be completed. Since the speed of convergence depends on the model and the actual systems parameters according to Equation (4), on the parameter $\alpha$ in Equation (9) or the interpolation parameter $\lambda_a$ used in [44], the general possibility to improve accuracy is the reduction in digital cycle time. In this manner, the number of the iterative steps made during unit time can be increased.

**Author Contributions:** Individual contributions by the Authors: Conceptualization, A.A.; Formal analysis, J.K.T.; Methodology, A.A.; Software, A.A.; Validation, J.K.T.; Writing—original draft, J.K.T. All authors have read and agreed to the published version of the manuscript.

**Funding:** This research was supported by the National Research, Development and Innovation Office (NKFIH) under OTKA Grant Agreement No. K 135512. We also acknowledge the support by the Doctoral School of Applied Informatics and Applied Mathematics of Óbuda University.

**Data Availability Statement:** Not applicable.

**Conflicts of Interest:** The authors declare no conflict of interest.

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
