# Peer review of "Tackling Modeling and Kinematic Inconsistencies by Fixed Point Iteration-Based Adaptive Control"

_machines, doi:10.3390/machines11060585_

Round 1

Reviewer 1 Report

In this paper, a fixed-point iteration-based adaptive control design is proposed to develop a solution of computed torque control for the robots. The topic is interesting, and there are some problems in the article should be fixed before it is published:

1. Please polish the language. Some sentences in the article are confusing that make the readers hard to understand the explanation and argument. For example, line 7 of the abstract.

2. The structure of the article should be rearranged. In my opinion, Introduction should clearly state the contribution of the research compared with existed result, the problem formulation and design structure are recommended to be given in the following section, and section II should be written as a subsection or paragraph to make the article compact and readable.

3. In figure 4, the structure of the coordinate system is confusing. Is that the normal Cartesian coordinate? The authors should clearly define the axes.

4. As the main result of the research, the derivation to get the control torque Q1 to Q3 in (5) should be given.

Reviewer 3 Report

In this work, the Fixed Point Iteration-based Adaptive Controller was systematically investigated for the control of approximately modeled underactuated systems. The paper is interesting. However, following concerns need to addressed.

1. Abstract is poorly written. It should clearly present the problem and significance of proposed solution with findings.

2. Only three keywords are defined. Please include one or two more keywords.

3. Literature review is not extensive. Please include recently published relevant articles and present how the proposed work is original.

4. Conclusion is very long. Please summarize findings and potential shortcomings or assumptions in the proposed work.

Round 2

Reviewer 3 Report

The paper can be accepted for publication.
